# EFFICIENT TRAINING OF MULTI-TASK COMBINAROTIAL NEURAL SOLVER WITH MULTI-ARMED BANDITS

## ABSTRACT

Efficiently training a multi-task neural solver for various combinatorial optimization problems (COPs) has been less studied so far. In this paper, we propose a general and efficient training paradigm based on multi-armed bandits to deliver a unified combinarotial multi-task neural solver. To this end, we resort to the theoretical loss decomposition for multiple tasks under an encoder-decoder framework, which enables more efficient training via proper bandit task-sampling algorithms through an intra-task influence matrix. Our method achieves much higher overall performance with either limited training budgets or the same training epochs, compared to standard training schedules, which can be promising for advising efficient training of other multi-task large models. Additionally, the influence matrix can provide empirical evidence of some common practices in the area of learning to optimize, which in turn supports the validity of our approach.

## 1 INTRODUCTION

Although a generic neural solver for multiple combinatorial optimization problems (COPs) is appealing, this problem is less studied in the literature, and training such a neural solver can be prohibitively expensive, especially in the era of large models. To relieve the training burden and better balance the resource allocation, in this paper, we propose a novel training paradigm via multi-armed bandits (MAB) from a multi-task learning (MTL) perspective, which can efficiently train a multi-task combinarotial neural solver under limited training budgets.

To this end, we treat each COP with a specific problem scale as a task and manage to deliver a generic solver handling a set of tasks simultaneously. Different from a standard joint training in MTL, we employ MAB algorithms to select/sample one task in each training round, hence avoiding the complex balancing of losses from multiple tasks. To better guide the MAB algorithms, we employ a reasonable reward design derived from the theoretical loss decomposition for the widely adopted encoder-decoder architecture in MTL. This loss decomposition also brings about an influence matrix revealing the mutual impacts between tasks, which provides rich evidence to explain some common practices in the scope of COPs.

To emphasize, our method is the first to consider training a generic neural solver for different kinds of COPs. This greatly differs from existing works focusing on either solution construction (Vinyals et al., 2015; Bello et al., 2017; Kool et al., 2019; Kwon et al., 2020) or heuristic improvement (Lu et al., 2020; Wu et al., 2021b; Agostinelli et al., 2021; Fu et al., 2021; Kool et al., 2022). Some recent works seek to generalize neural solvers to different scales (Hou et al.; Li et al., 2021; Cheng et al., 2023; Wang et al., 2023) or varying distributions (Wang et al., 2021; Bi et al., 2022; Geisler et al., 2022), but with no ability to handle multiple types of COPs simultaneously.

Experiments are conducted for 12 tasks: Four types of COPs, the Travelling Salesman Problem (TSP), the Capacitated Vehicle Routing Problem (CVRP), the Orienteering Problem (OP) and the Knapsack Problem (KP), and each of them with three problem scales. We compare our approach with single-task training (STL) and extensive MTL baselines (Mao et al., 2021; Yu et al., 2020; Navon et al., 2022; Kendall et al., 2018; Liu et al., 2021a;b) under the cases of the same training budgets and same training epochs. Compared with STL, our approach needs no prior knowledge about tasks and can automatically focus on harder tasks so as to maximally utilize the training budget. What's more, when comparing with STL under the same training epoch, our approach not only enjoys the cheaper training cost which is strictly smaller than that of the most expensive task, but also shows the

generalization ability by providing a universal model to cover different types of COPs. Compared with the MTL methods, our method only picks the most impacting task to train at each time which improves the training efficiency without explicitly balancing the losses.

In summary, our contributions can be concluded as follows: **(1)** We propose a novel framework for efficiently training a combinatorial neural solver for multiple COPs via MAB, which achieves prominent performance against standard training paradigms with limited training resources and can further advise efficient training of other large models; **(2)** We study the theoretical loss decomposition for the encoder-decoder architecture, leading to the influence matrix reflecting the inherent task relations and reasonable reward guiding the update of MAB algorithms.; **(3)** We verify several empirical observations for neural solvers from previous works (Kool et al., 2019; Joshi et al., 2021) by the influence matrix, demonstrating the validity and reasonableness of our approach.

## 2 RELATED WORK

**Neural solvers for COPs.** Pointer Networks (Vinyals et al., 2015) pioneered the application of deep neural networks for solving combinatorial optimization problems. Subsequently, numerous neural solvers have been developed to address various COPs, such as routing problems (Bello et al., 2017; Kool et al., 2019; Lu et al., 2020; Wu et al., 2021b;b), knapsack problem (Bello et al., 2017; Kwon et al., 2020), job shop scheduling problem (Zhang et al., 2020), and others. There are two prevalent approaches to constructing neural solvers: solution construction (Vinyals et al., 2015; Bello et al., 2017; Kool et al., 2019; Kwon et al., 2020), which sequentially constructs a feasible solution, and heuristic improvement (Lu et al., 2020; Wu et al., 2021b; Agostinelli et al., 2021; Fu et al., 2021; Kool et al., 2022), which provides meaningful information to guide downstream classical heuristic methods. In addition to developing novel techniques, several works (Wang et al., 2021; Geisler et al., 2022; Bi et al., 2022; Wang et al., 2023) have been proposed to address generalization issues inherent in COPs. For a comprehensive review of the existing challenges in this area, we refer to the survey (Bengio et al., 2020).

**Multi-task learning.** Multi-Task Learning (MTL) aims to enhance the performance of multiple tasks by jointly training a single model to extract shared knowledge among them. Numerous works have emerged to address MTL from various perspectives, such as exploring the balance on the losses from different tasks (Mao et al., 2021; Yu et al., 2020; Navon et al., 2022; Kendall et al., 2018; Liu et al., 2021a;b) designing module-sharing mechanisms (Misra et al., 2016; Sun et al., 2020; Hu & Singh, 2021), improving MTL through multi-objective optimization (Sener & Koltun, 2018; Lin et al., 2019; Momma et al., 2022), and meta-learning (Song et al., 2022). To optimize MTL efficiency and mitigate the impact of negative transfer, some research focuses on task-grouping (Kumar & III, 2012; Zamir et al., 2018; Standley et al., 2020; Fifty et al., 2021), with the goal of identifying task relationships and learning within groups to alleviate negative transfer effects in conflicting tasks. On the application level, MTL has been extensively employed in various domains, including natural language processing (Collobert & Weston, 2008; Luong et al., 2016), computer vision (Zamir et al., 2018; Seong et al., 2019), bioinformatics Xu et al. (2017), and many others. However, there are limited works on solving COPs using MTL. In this work, we highlight research on MTL for COPs and propose a learning framework to concurrently address various types of COPs.

**Multi-armed bandits.** Multi-armed bandit (MAB) is a classical problem in decision theory and machine learning that addresses the exploration-exploitation trade-off. Several algorithms and strategies have been suggested to solve the MAB problem, such as the $\epsilon$-greedy, Upper Confidence Bound (UCB) family of algorithms (Lai et al., 1985; Auer et al., 2002), the Exp3 family (Littlestone & Warmuth, 1994; Auer et al., 1995; Gur et al., 2014), and the Thompson sampling (Thompson, 1933; Agrawal & Goyal, 2012; Chapelle & Li, 2011). These methods differ in their balance of exploration and exploitation, and their resilience under distinct types of uncertainty. The MAB has been extensively studied in both theoretical and practical contexts, and comprehensive details can be found in Slivkins et al. (2019); Lattimore & Szepesvári (2020).

## 3 METHOD

We consider $K$ types of COPs, denoted as $T^i$ $(i = 1, 2, ..., K)$, with $n_i$ different problem scales for each COP. Thus, the overall task set is $\mathcal{T} = \bigcup_{i=1}^{K} T^i := \{T_j^i | j = 1, 2, ..., n_i, i = 1, 2, ..., K\}$.

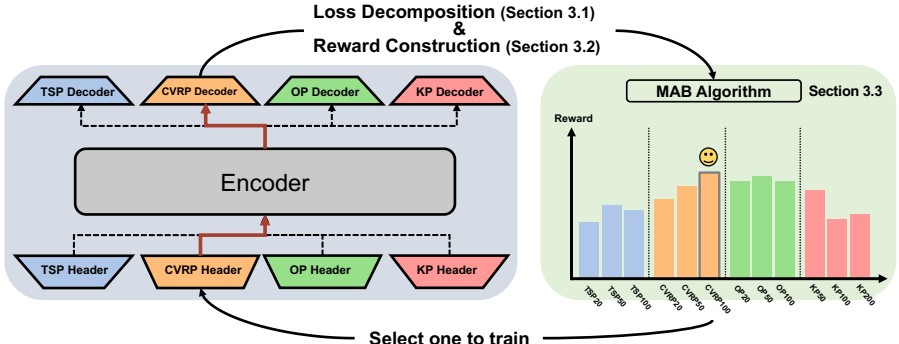

Figure 1: Pipeline of MAB for Solving COPs in view of MTL. We consider four types of COPs: TSP, CVRP, OP and KP, each with a corresponding header and decoder. The encoder, which is common to all COPs, is also included. For each time step, we utilize the MAB algorithm to select a specific task for training, such as CVRP-100 depicted in the figure. We then obtain the loss for the selected task, perform loss decomposition as detailed in Section 3.1, and construct a reward using the methodology outlined in Section 3.2. Finally, we utilize the reward to update the MAB algorithm.

---

**Algorithm 1** MAB for Solving COPs in view of MTL

---

**Require:** Combinatorial neural solver $S_\Theta$ with parameters $\Theta$, task set $\mathcal{T}$, MAB algorithm $\mathcal{A}(\mathcal{T})$, loss function $L(\Theta)$, number of training loops $L$, update frequency for MAB algorithm $freq$.
1: **for** $t = 1$ **to** $L$ **do**
2:  Train $S_{\Theta(t)}$ on task $T_j^i$ selected by $\mathcal{A}(\mathcal{T})$ and store the gradient information $\nabla L_j^i(\Theta(t))$
3:  **if** $t \bmod freq = 0$ **then**
4:   Obtaining reward $\bar{r}_j^i$ for each task $T_j^i$ using stored gradients $\{\nabla L_j^i(\Theta(t))\}_{t=t_1}^{t_2}$ following Section 3.2
5:   Update $\mathcal{A}(\mathcal{T})$ with reward $\bar{r}_j^i$ for each task $T_j^i$
6:   Clear the record of the gradient information
7:  **end if**
8: **end for**
9: **return** Well-trained neural solver $S_\Theta$

---

For each type of COP $T^i$, we consider a neural solver $S_{\Theta^i}(\mathcal{I}_j^i) : T_j^i \to \mathcal{Y}_j^i$, where $\Theta^i$ are the parameters for COP $T^i$, $\mathcal{I}_j^i$ and $\mathcal{Y}_j^i$ are the input instance the output space for COP $T^i$ with the problem scale of $n_j$ (termed as task $T_j^i$ in the sequel). The parameter vector $\Theta^i = (\theta^{\text{share}}, \theta^i)$ contains the shared and task-specific parameters for the COP $T^i$, and the complete set of parameters is denoted by $\Theta = \bigcup_{i=1}^{K} \Theta^i$. This parameter notation corresponds to the commonly used Encoder-Decoder framework [1] in multi-task learning in Fig. 1, where $\theta^{\text{share}}$ represents the encoder - shared across all tasks, and $\theta^i$ represents the decoder - task-specific for each task. Given the task loss functions $L_j^i(\Theta^i)$ for COP $T^i$ with the problem scale of $n_j$, we investigate the widely used objective function:

$$\min_{\Theta} L(\Theta) = \sum_{i=1}^{K} \sum_{j=1}^{n_i} L_j^i(\Theta^i). \tag{1}$$

We propose a general framework based on Multi-Armed Bandits (MAB) to dynamically select tasks during training rounds and a reasonable reward is constructed to guide the selection process. In particular, our approach establishes a comprehensive task relation by the obtained influence matrix, which has the potential to empirically validate several common deep learning practices while solving COPs.

**Overview**. We aim to solve Eq. 1 using the MAB approach. Given the set of tasks $\mathcal{T} = \{T_j^i | j = 1, 2, ..., n_i, i = 1, 2, ..., K\}$, we select an arm (i.e., task being trained) $a_t \in \mathcal{T}$ following an MAB algorithm, which yields a random reward signal $r_t$ that reflects the effect of the selection. The approximated expected reward is updated based on the received rewards. Essentially, our proposed

---

[1]According to the Encoder-Decoder framework, encoder commonly refers to shared models, whereas decoder concerns task-specific modules. In this study, the decoder component comprises two modules: "Header" and "Decoder" as illustrated in Figure 1.

method is applicable to any MAB algorithm. The general framework of MAB for solving COPs within the context of Multi-Task Learning (MTL) is outlined in Algorithm 1, and the overall pipeline is illustrated in Figure 1.

## 3.1 LOSS DECOMPOSITION

In the framework of MAB for solving COPs in view of MTL described in Algorithm 1, the way to design a reasonable reward to guide its update is crucial. In this part, we analytically drive a reasonable reward by decomposing the loss function for the Encoder-Decoder framework in Fig. 1. Following the previous notation, $\Theta = \bigcup_{i=1}^{K} \Theta^i = \{\theta^{\text{share}}\} \bigcup \{\theta_i, i = 1, 2, ..., K\}$ are all trainable parameters.

We suppose that a meaningful reward should satisfy the following two properties: **(1)** It can benefit our objective and reveal the intrinsic training signal; **(2)** When a task is selected, there always has positive effects on it in expectation.

The difference on loss function is an ideal choice and previous work has used it to measure the task relationship (Fifty et al., 2021). However, such measurement is invalid in our context because there are no significant differences among tasks (see Appendix F), so using such information may mislead the bandit selection. What's more, the computation cost of the *"lookahead loss"* in Fifty et al. (2021) is considerably expensive when frequent reward signals are needed. We instead propose a more fundamental way based on gradients to measure the impacts of training one task upon the others.

To simplify the analysis, in Proposition 1 we assume the standard gradient descent (GD) is used to optimize Eq. 1 by training one task at each step $t$, and then derive the loss decomposition under the encoder-decoder framework. Any other optimization method, e.g., Adam (Kingma & Ba, 2015), can also be used here with small modifications. We leave the detailed proofs for GD and Adam optimizer in Appendix B.

**Proposition 1** (Loss decomposition for GD). *Using encoder-decoder framework with parameters* $\Theta = \bigcup_{i=1}^{K} \Theta^i = \{\theta^{share}\} \bigcup \{\theta_i, i = 1, 2, ..., K\}$ *and updating parameters with standard gradient descent:* $\Theta(t+1) = \Theta(t) - \eta_t \nabla L(\Theta(t))$, *where* $\eta_t$ *is the step size. Then the difference of the loss of task* $T_j^i$ *from training step* $t_1$ *to* $t_2$: $\Delta L_j^i(t_1 \to t_2) = L_j^i(\Theta^i(t_2)) - L_j^i(\Theta^i(t_1))$ *can be decomposed to:*

$$\Delta L_j^i(t_1 \to t_2)$$

$$= - \underbrace{(\nabla^T L_j^i(\Psi^i(t_1)) \sum_{t=t_1}^{t_2} \mathbb{1}(a_t = T_j^i) \eta_t \nabla L_j^i(\Theta^i(t))}_{(a)\ \textit{effects of training task } T_j^i:\ e_j^i(t_1 \to t_2)} + \underbrace{\nabla^T L_j^i(\Psi^i(t_1)) \sum_{\substack{q=1 \\ q \neq j}}^{n_i} \sum_{t=t_1}^{t_2} \mathbb{1}(a_t = T_q^i) \eta_t \nabla L_q^i(\Theta^i(t))}_{(b)\ \textit{effects of training task } \{T_q^i, q \neq j\}:\ \{e_q^i((t_1 \to t_2)), q \neq j\}}$$

$$+ \underbrace{\nabla_{\theta^{share}}^T L_j^i(\Psi^i(t_1)) \sum_{\substack{p=1 \\ p \neq i}}^{K} \sum_{q=1}^{n_p} \sum_{t=t_1}^{t_2} \mathbb{1}(a_t = T_q^p) \eta_t \nabla_{\theta^{share}}^T L_q^p(\Theta^p(t)))}_{(c)\ \textit{effects of training task } \{T_q^p, p \neq i\}:\{e_q^p(t_1 \to t_2), q=1,2,...,n_p, p \neq i\}},$$

$$(2)$$

*where* $\nabla L(\Theta)$ *means taking gradient w.r.t.* $\Theta$ *and* $\nabla L_\theta(\Theta)$ *means taking gradient w.r.t.* $\theta \subseteq \Theta$, $\Psi^i(t_1)$ *is some vector between* $\Theta^i(t_1)$ *and* $\Theta^i(t_2)$ *and* $\mathbb{1}(a_t = T_j^i)$ *is the indicator function.*

The idea behind Eq. 2 means the improvement on the loss for task $T_j^i$ from $t_1$ to $t_2$ can be decomposed into three parts: (**a**) effects of training $T_j^i$ itself w.r.t. $\Theta^i$; (**b**) effects of training same kind of COP $\{T_q^i, q \neq j\}$ w.r.t. $\Theta^i$; and (**c**) effects of training other COPs $\{T^p, p \neq i\}$ w.r.t. $\theta^{\text{share}}$. Indeed, we quantify the impact of different tasks on $T_j^i$ through this decomposition, which provides the intrinsic training signals for designing reasonable rewards.

## 3.2 REWARD DESIGN AND INFLUENCE MATRIX CONSTRUCTION

In this part, we design the reward and construct the intra-task relations based on the loss decomposition introduced in Section 3.1. Though Eq. 2 reveals the signal during training, the inner products of

gradients from different tasks can significantly differ at scale (see Appendix F). This will mislead the bandit's update seriously since improvements may come from large gradient values even when they are almost orthogonal. To address this, we propose to use cosine metric to measure the influence between task pairs. Formally, for task $T_j^i$ from $t_1$ to $t_2$, the influence from training the same type of COP $T_q^i$ to $T_j^i$ is:

$$m_q^i(t_1 \to t_2) = \frac{\nabla^T L_j^i(\Psi^i(t_1)) \sum_{t=t_1}^{t_2} \eta_t \mathbb{1}(a_t = T_q^i) \nabla L_q^i(\Theta^i(t))}{||\nabla^T L_j^i(\Psi^i(t_1))|| \cdot ||\sum_{t=t_1}^{t_2} \eta_t \mathbb{1}(a_t = T_q^i) \nabla L_q^i(\Theta^i(t))||}, \tag{3}$$

and the influence from training other types of COPs $T_q^p$ to $T_j^i$ is:

$$m_q^p(t_1 \to t_2) = \frac{\nabla_{\theta^{\text{share}}}^T L_j^i(\Psi^i(t_1)) \sum_{t=t_1}^{t_2} \eta_t \mathbb{1}(a_t = T_q^p) \nabla_{\theta^{\text{share}}} L_q^p(\Theta^p(t))}{||\nabla_{\theta^{\text{share}}}^T L_j^i(\Psi^i(t_1))|| \cdot ||\sum_{t=t_1}^{t_2} \eta_t \mathbb{1}(a_t = T_q^p) \nabla_{\theta^{\text{share}}} L_q^p(\Theta^p(t))||}. \tag{4}$$

Given Eq. 3, 4, we denote the influence vector to $T_j^i$ as:

$$\mathbf{m}_j^i(t_1 \to t_2) = (..., \underbrace{m_1^i(t_1 \to t_2), ..., \overbrace{m_j^i(t_1 \to t_2)}^{\text{influence from itself}}, ..., m_{n_i}^i(t_1 \to t_2)}_{\text{influence from the same kind of COP}}, \underbrace{..., m_q^p(t_1 \to t_2), ...}_{\text{influence from other kinds of COPs}})^T \tag{5}$$

Based on Eq. 5, an influence matrix $M(t_1 \to t_2) = (..., \mathbf{m}_j^i(t_1 \to t_2), ...)^T \in \mathbb{R}^{\sum_{k=1}^K n_k \times \sum_{k=1}^K n_k}$ can be constructed to reveal the relationship between tasks from time step $t_1$ to $t_2$. There are several properties about influence matrix $M(t_1 \to t_2)$: **(1)** $M(t_1 \to t_2)$ has blocks $M^i(t_1 \to t_2) \in \mathbb{R}^{n_i}$ in the diagonal position which is the sub-influence matrix of a same kind of COP with different problem scales; **(2)** $M(t_1 \to t_2)$ is asymmetry which is consistent with the general understanding in multi-task learning; **(3)** The row-sum of $M(t_1 \to t_2)$ are the total influences obtained *from* all tasks *to* one task; **(4)** The column-sum of $M(t_1 \to t_2)$ are the total influences *from* one task *to* all tasks.

According to the implication of the elements in $M(t_1 \to t_2)$, the column-sum of $M(t_1 \to t_2)$:

$$\mathbf{r}(t_1 \to t_2) = \mathbf{1}^T \cdot M(t_1 \to t_2) \in \mathbb{R}^{1 \times \sum_{k=1}^K n_k} \tag{6}$$

actually provides a meaningful reward signal for selecting tasks ,which we can use to update the bandit algorithm. Moreover, we denote the update frequency of computing the influence matrix as $\Delta T$ and the overall training time is $n\Delta T$, then an average influence matrix $W$ can be constructed based on influence matrices $\{M(k\Delta T \to (k+1)\Delta T), k = 0, 2, ..., n-1\}$ collected during the training process:

$$W = \frac{1}{n\Delta T} \sum_{k=1}^{n-1} M(k\Delta T \to (k+1)\Delta T), \tag{7}$$

revealing the overall task relations across the training process.

When computing the bandit rewards, there remains an issue regarding the approximation of $\nabla^T L_j^i(\Psi^i(t_1))$ in equations 3 and 4. Moreover, there is a lack of theoretical works discussing this issue within the context of neural networks. We propose a heuristic method that relies on the widely accepted assumption in multi-task learning:

**Assumption 1.** *When using cosine metric on the gradients to measure the similarity between tasks, one task should have the similarity of 1 with itself (Wang et al., 2020; Yu et al., 2020).*

The training influences determined by Eq. 3 and 4 can be seen as the similarity between tasks measured by cosine metric, therefore we can determine:

$$\nabla^T L_j^i(\Psi^i(t_1)) = \sum_{t=t_1}^{t_2} \eta_t \mathbb{1}(a_t = T_j^i) \nabla L_j^i(\Theta^i(t)) \tag{8}$$

for Eq. 3 when $q = j$ in order to ensure that the self-task similarity $m_j^i(t_1 \to t_2)$ equals 1.

## 4 EXPERIMENTS

In this section, we conduct a comparative analysis between our proposed method and both single-task training (STL) and extensive multi-task learning (MTL) methods to demonstrate the efficacy of our approach in addressing various COPs under different evaluation criteria. Specifically, we examine two distinct scenarios: **(1)** Under identical training budgets, we aim to showcase the convenience of our method in automatically obtaining a universal combinatorial neural solver for multiple COPs, circumventing the challenges of balancing loss in MTL and allocating time for each task in STL; **(2)** Given the same number of training epochs, we seek to illustrate that our method can derive a potent neural solver with excellent generalization capability. Furthermore, we employ the influence matrix to analyze the relationship between different COP types and the same COP type with varying problem scales.

**Experimental settings**. We explore four types of COPs: the Travelling Salesman Problem (TSP), the Capacitated Vehicle Routing Problem (CVRP), the Orienteering Problem (OP), and the Knapsack Problem (KP). Detailed descriptions can be found in Appendix A. Three problem scales are considered for each COP: 20, 50, and 100 for TSP, CVRP, and OP; and 50, 100, and 200 for KP. We employ the notation "COP-scale", such as TSP-20, to denote a particular task, resulting in a total of 12 tasks. We emphasize that the derivation presented in Section 3.1 applies to a wide range of loss functions encompassing both supervised learning-based and reinforcement learning-based methods. In this study, we opt for reinforcement learning-based neural solvers, primarily because they do not necessitate manual labeling of high-quality solutions. As a representative method in this domain, we utilize the Attention Model (AM) (Kool et al., 2019) as the backbone and employ POMO (Kwon et al., 2020) to optimize its parameters. Concerning the bandit algorithm, we select Exp3 and the update frequency is set to 12 training batches. We discuss the selection of the MAB algorithms and update frequency in Appendix C, with details on training and configuration in Appendix E.

### 4.1 COMPARISON WITH SINGLE TASK TRAINING AND MULTI TASK LEARNING

In this part, we explore the differences in performance between our method, MTL, and STL across various comparison criteria, highlighting our method's superior efficiency and generalization ability.

**Comparison under same training budgets.** We now consider a practical scenario with limited training resources available for neural solvers for all tasks. Our method addresses this challenge by concurrently training all tasks using an appropriate task sampling strategy. However, establishing a schedule for STL is difficult due to the lack of information regarding resource allocation for each task, and MTL methods are hindered by efficiency issues arising from joint task training. In this section, we compare our method with naive STL and MTL methods in terms of the optimality gap: $gap\% = |\frac{\text{obj.}}{\text{gt.}} - 1| \times 100$, averaged over 10,000 instances for each task under an identical training time budget.

Table 1: Training time per epoch, represented in minutes. The COPs are classified into three scales: small, median, and large, which correspond to the sizes of 20, 50, and 100, respectively (50, 100, and 200 for KP).

| COP | Small | Median | Large |
|------|-------|--------|-------|
| TSP | 0.19 | 0.39 | 0.75 |
| CVRP | 0.27 | 0.50 | 0.90 |
| OP | 0.20 | 0.41 | 0.60 |
| KP | 0.34 | 0.61 | 1.10 |

The total training time budget is designated as $B$, with each type of COP receiving resources equitably for $\frac{B}{4}$ within the STL framework. Two schedules are considered for the allocation of time across varying problem scales for the same category of COP: **(1)** Average allocation, denoted as $\text{STL}_{\text{avg.}}$, indicating a uniform distribution of resources for each task; **(2)** Balanced allocation, denoted as $\text{STL}_{\text{bal.}}$, signifying a size-dependent resource assignment with a 1:2:3 ratio from small to large problem scales, categorizing tasks into easy-median-hard levels. The first schedule is suitable for realistic scenarios where information regarding the tasks is unavailable, while the second is advantageous when prior knowledge is introduced.

To mitigate the impact of extraneous computations, we calculate the time necessary to complete one epoch for each task and convert the training duration into the number of training epochs for STL. Utilizing the same device, the training time for for each task with STL and MTL methods can be found in Table 1 and Table 3. We assess three distinct training budgets: **(1)** Small budget: the time required to complete 500 training epochs using our method, approximately 1.59 days in GPU hours;

Table 2: Comparison among our proposed method, multi-task learning (MTL), and single task training (STL) utilizing the same training budget. Specifically, STLavg. and STLbal. denote the allocation of resources, with an even distribution and a balanced allocation ratio of $1:2:3$, respectively, among tasks with varying scales from small to large. The reported results depict the optimality gap ($\downarrow$) in the main aspects.

| | Method | TSP20 | TSP50 | TSP100 | CVRP20 | CVRP50 | CVRP100 | OP20 | OP50 | OP100 | KP50 | KP100 | KP200 | Avg. Gap |
|---|---|---|---|---|---|---|---|---|---|---|---|---|---|---|
| Small Budget | STLavg. | **0.009%** | 0.346% | 3.934% | 0.465% | 2.292% | 5.899% | −1.075% | 1.291% | 5.674% | **0.029%** | 0.015% | 0.017% | 1.575% |
| | STLbal. | 0.019% | 0.346% | 2.967% | 0.599% | 2.292% | 4.774% | −0.973% | 1.291% | 4.771% | 0.033% | 0.015% | 0.016% | 1.346% |
| | Naive-MTL | 0.029% | 0.725% | 3.427% | 0.676% | 2.455% | 4.396% | −0.464% | 2.607% | 5.564% | 0.036% | 0.014% | 0.016% | 1.623% |
| | Bandit-MTL | 0.023% | 0.804% | 3.601% | 0.717% | 2.523% | 4.460% | −0.715% | 1.148% | 2.903% | 0.047% | 0.016% | 0.021% | 1.296% |
| | PCGrad | 0.230% | 1.762% | 5.476% | 1.337% | 4.025% | 6.858% | 0.323% | 4.421% | 7.773% | 64.504% | 0.018% | 0.036% | 8.064% |
| | UW | 0.036% | 0.394% | 1.905% | 0.451% | 1.667% | 3.291% | −0.562% | 1.776% | 3.989% | 0.039% | 0.016% | 0.022% | 1.085% |
| | CAGrad | 0.634% | 3.209% | 8.433% | 1.417% | 4.631% | 7.668% | 0.536% | 4.516% | 8.232% | 0.048% | 0.024% | 0.063% | 3.284% |
| | IMTL | 27.539% | 53.406% | 77.085% | 175.989% | 345.701% | 560.506% | 8.620% | 31.643% | 52.342% | 46.968% | 53.615% | 71.868% | 125.440% |
| | Nash-MTL | 0.274% | 1.315% | 4.286% | 0.858% | 3.166% | 5.852% | −0.171% | 3.494% | 7.312% | 0.045% | 0.016% | 0.021% | 2.206% |
| | Random | 0.041% | 0.402% | 1.975% | 0.489% | 1.797% | 3.298% | −0.998% | 0.794% | 2.488% | 0.032% | 0.014% | 0.015% | 0.862% |
| | Ours | 0.030% | **0.297%** | **1.687%** | **0.422%** | **1.554%** | **2.861%** | **−1.081%** | **0.533%** | **2.153%** | 0.031% | 0.014% | **0.014%** | **0.710%** |
| Medium Budget | STLavg. | **0.005%** | **0.183%** | 2.237% | 0.379% | 1.667% | 4.137% | **−1.156%** | 0.674% | 3.670% | **0.025%** | 0.013% | 0.015% | 0.988% |
| | STLbal. | 0.008% | 0.183% | 1.656% | 0.447% | 1.667% | 3.460% | −1.118% | 0.674% | 2.563% | 0.027% | 0.013% | 0.013% | 0.800% |
| | Naive-MTL | 0.023% | 0.376% | 2.058% | 0.507% | 1.987% | 3.576% | −0.831% | 1.277% | 3.416% | 0.032% | **0.012%** | 0.013% | 1.037% |
| | Bandit-MTL | 0.015% | 0.424% | 2.072% | 0.586% | 1.857% | 3.483% | −1.137% | 0.805% | 2.577% | 0.047% | 0.016% | 0.021% | 0.897% |
| | PCGrad | 0.097% | 1.070% | 3.941% | 0.897% | 3.003% | 5.016% | −0.100% | 3.142% | 5.782% | 64.504% | 0.019% | 0.018% | 7.282% |
| | UW | 0.036% | 0.231% | 1.426% | 0.373% | 1.540% | 3.032% | −0.855% | 0.802% | 3.108% | 0.033% | 0.013% | 0.022% | 0.813% |
| | CAGrad | 0.507% | 2.069% | 6.112% | 1.353% | 3.489% | 6.113% | −0.176% | 3.800% | 6.957% | 0.044% | 0.024% | 0.063% | 2.530% |
| | IMTL | 27.539% | 53.406% | 77.085% | 20.549% | 57.972% | 117.359% | 5.431% | 24.835% | 38.553% | 3.700% | 3.986% | 1.973% | 36.032% |
| | Nash-MTL | 0.157% | 0.854% | 3.104% | 0.617% | 2.415% | 4.561% | −0.677% | 2.679% | 5.596% | 0.045% | 0.016% | 0.021% | 1.616% |
| | Random | 0.025% | 0.252% | 1.310% | 0.371% | 1.438% | 2.608% | −1.099% | 0.339% | 1.583% | 0.029% | 0.013% | 0.014% | 0.574% |
| | Ours | 0.019% | 0.202% | **1.086%** | **0.348%** | **1.284%** | **2.362%** | −1.114% | **0.224%** | **1.277%** | 0.030% | **0.012%** | **0.012%** | **0.478%** |
| Large Budget | STLavg. | **0.002%** | **0.114%** | 1.296% | **0.282%** | 1.276% | 3.071% | **−1.223%** | 0.253% | 2.087% | **0.018%** | 0.011% | 0.014% | 0.600% |
| | STLbal. | 0.005% | 0.114% | 1.024% | 0.367% | 1.276% | 2.601% | −1.175% | 0.253% | 1.609% | 0.024% | 0.011% | 0.012% | 0.510% |
| | Naive-MTL | 0.015% | 0.236% | 1.343% | 0.393% | 1.589% | 2.920% | −0.971% | 0.504% | 1.919% | 0.026% | 0.012% | 0.012% | 0.667% |
| | Bandit-MTL | 0.012% | 0.261% | 1.479% | 0.450% | 1.529% | 2.927% | −1.166% | 0.419% | 1.928% | 0.036% | 0.013% | 0.017% | 0.659% |
| | PCGrad | 0.036% | 0.674% | 2.948% | 0.589% | 2.311% | 3.945% | −0.764% | 1.836% | 4.056% | 0.039% | 0.015% | 0.018% | 1.309% |
| | UW | 0.025% | 0.195% | 1.230% | 0.347% | 1.399% | 2.718% | −0.855% | 0.588% | 2.022% | 0.028% | 0.013% | 0.015% | 0.644% |
| | CAGrad | 0.347% | 1.955% | 5.822% | 1.353% | 2.723% | 4.810% | −0.343% | 2.501% | 5.001% | 0.056% | 0.024% | 0.026% | 2.067% |
| | IMTL | 27.539% | 53.406% | 77.085% | 10.597% | 23.232% | 33.583% | 3.953% | 25.055% | 39.207% | 0.059% | 0.030% | 0.031% | 24.481% |
| | Nash-MTL | 0.079% | 0.513% | 2.331% | 0.517% | 2.191% | 3.589% | −0.948% | 1.437% | 3.996% | 0.045% | 0.018% | 0.021% | 1.149% |
| | Random | 0.019% | 0.207% | 1.069% | 0.336% | 1.274% | 2.322% | −1.104% | 0.219% | 1.188% | 0.027% | 0.012% | 0.012% | 0.465% |
| | Ours | 0.017% | 0.169% | **0.960%** | 0.335% | **1.218%** | **2.236%** | −1.139% | **0.149%** | **1.129%** | 0.026% | **0.011%** | **0.011%** | **0.427%** |

**(2)** Medium budget: 1000 training epochs, consuming 3.28 days in GPU hours; and **(3)** Large budget: 2000 training epochs, spanning 6.64 days in GPU hours.

Extensive MTL baselines are considered here: Bandit-MTL (Mao et al., 2021), PCGrad (Yu et al., 2020), Nash-MTL (Navon et al., 2022), Uncertainty-Weighting (UW) (Kendall et al., 2018), CAGrad (Liu et al., 2021a) and IMTL (Liu et al., 2021b). We also involve the random policy which samples the task uniformly at each training slot, and the results are presented in Table 2. In general, our method outperforms MTL and STL methods in terms of averge gap across all the budgets used. Specifically, our method yields consistent improvements for 10 out of 12 tasks under the small budget, 8 and 7 out of 12 tasks under the medium and large budget. Moreover, our approach demonstrates a stronger focus on more challenging problems, as it attains greater improvements for larger problem scales compared to smaller ones. What's more, when comparing with all MTL methods, our method demonstrates two superior advantages:

- Better performance on the solution quality and efficiency: In Table 2, typical MTL methods fail to obtain a powerful neural solver efficiently, and some of them even work worse than naive MTL and STL in limited budgets;
- More resources-friendly: The computation complexity of typical MTL methods grows linearly w.r.t. the number of tasks [2], conducting these training methods still needs heavy training resources (High-performance GPU with quite large memories). The exact training time for one epoch w.r.t. GPU hour are listed in Table 3. Under the same training setting, intermediate termination of prolonged training epoch for typical MTL methods incurs wasted computation resources. However, our method trains only one task at each time slot, resulting in rapid epoch-wise training that facilitates flexible experimentation and iteration.

Table 3: Time consumption for MTL methods w.r.t. the GPU hours for training one epoch in average.

| | Bandit-MTL | PCGrad | Nash-MTL | UW | IMTL | CAGrad | Ours |
|---|---|---|---|---|---|---|---|
| GPU Hours | 1.04 | 6.02 | 5.87 | 1.00 | 5.61 | 5.24 | 0.07 |

It's also interesting to see that the random policy outperforms STL and the best-performing MTL baselines in our context, underscoring the positive effects of changing the training paradigm. Fur-

---

[2]Detailed analysis about the computation complexity of each MTL method is in Appendix D.

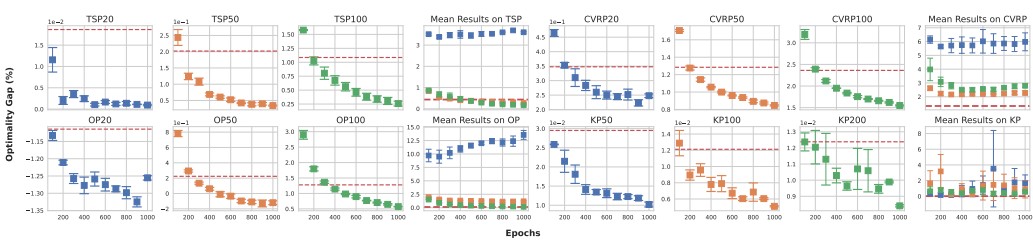

Figure 2: A comparison between single task training (STL) and our method is showcased in this figure, with both methods utilizing the same number of training epochs (1000 in this case). While STL achieves superior performance, our method is capable of effectively tackling all tasks simultaneously, as evidenced by the strong mean results it produces.

thermore, our proposed method surpasses the random policy, providing evidence of the additional improvements achieved through the integration of the bandit algorithm.

As the training budgets increase, STL's advantages become evident in easier tasks such as TSP, CVRP-20, OP-20, and KP-50. However, our method continues to deliver robust results for more difficult tasks like CVRP-100 and OP-100. Simultaneously, we observe a decrease in gain as the budget expands, aligning with our understanding that negative transfer exists among different tasks.

In addition to performance gains, the most notable advantage of our approach is that it does not require prior knowledge of the tasks and is capable of dynamically allocating resources for each task, which is crucial in real-world scenarios. When implementing STL, biases are inevitably introduced with equal allocation. As demonstrated in Table 2, the performance of two distinct allocation schedules can differ significantly: $STL_{bal.}$ consistently outperforms $STL_{avg.}$ due to the introduction of appropriate priors for STL.

Table 4: The comparison results are obtained by training our model for 1000 epochs and STL models for 100 epochs each, amounting to a total of 1200 epochs.

| | TSP20 | TSP50 | TSP100 | CVRP20 | CVRP50 | CVRP100 | OP20 | OP50 | OP100 | KP50 | KP100 | KP200 | Avg. Gap |
|---|---|---|---|---|---|---|---|---|---|---|---|---|---|
| STL | **0.011%** | 0.244% | 1.578% | 0.465% | 1.706% | 3.194% | **-1.133%** | 0.781% | 2.898% | **0.026%** | 0.013% | 0.01237% | 0.816% |
| Ours | 0.019% | **0.202%** | **1.086%** | **0.348%** | **1.284%** | **2.362%** | -1.114% | **0.224%** | **1.277%** | 0.030% | **0.012%** | **0.01236%** | **0.478%** |

**Comparison under same training epochs.** We conduct a comparison under the same number of training epochs by training our method on 12 tasks mentioned before for 1000 epochs in total, and comparing them with corresponding Single Task Learning (STL) neural solvers that are trained for 1000 epochs on each of their respective tasks. This is, by no means, a fair comparison, as our method dynamically chooses a task to train for 1000 epochs, resulting in a much smaller sample size than each task when using STL. Despite this, we choose this comparison as an intuitive way to demonstrate the superior generalization ability of our method under such extreme conditions. We present the results in Figure 2 and Table 4. Compared to individual tasks, shown in Table 4, our method (trained 1000 epochs) consistently outperforms STL (trained $100 \times 12 = 1200$ epochs) across most tasks, with exceptions noted in TSP20, OP20, and KP50. In most cases, our method's performance is equivalent to that of using 100 to 300 epochs of STL. However, STL can only obtain one model in this context and lacks the ability to handle different types of COPs or to generalize well when presented with the same type of COP but with varying problem scales. As a result, our method demonstrates unparalleled superiority in three ways: **(1)** when considering the average performance on all problem scales for each type of COP, our method obtains the best results in CVRP, OP, and KP, and is equivalent to the results achieved by training TSP for about 500 epochs. This showcases our method's excellent generalization ability for problem scales; **(2)** Our method can handle various types of COPs under the same number of training epochs, which is impossible for STL due to the existence of task-specific modules; **(3)** Our method's training time is strictly shorter than the longest time-consuming task.

## 4.2 STUDY OF THE INFLUENCE MATRIX

Our approach has an additional advantage as it facilitates the identification of the task relationship through the influence matrix developed in Section 3.2. The influence matrix allows us to capture the inherent relationship among tasks. Additionally, we provide empirical evidence pertaining to the experience and observation in the learning to optimize community. We present a detailed view of the influence matrix in Figure 3, revealing significant observations: **(1)** Figure 3a highlights that the

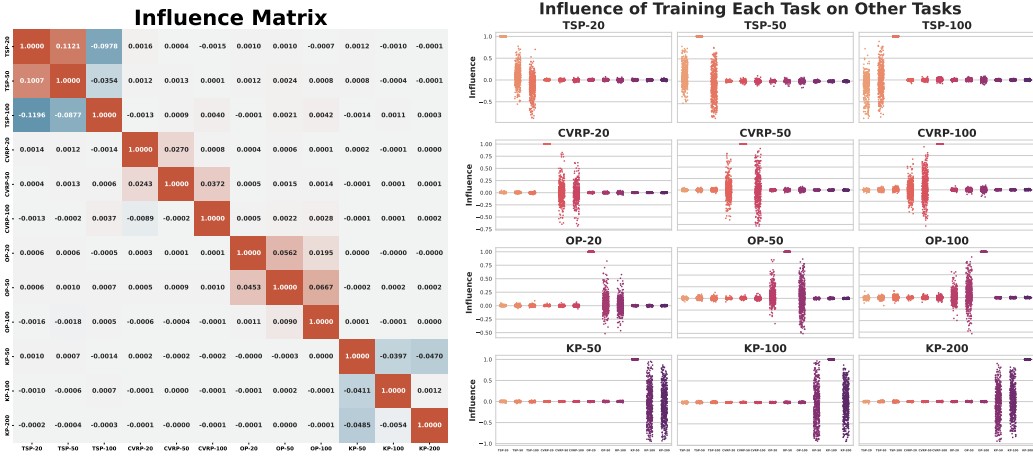

(a) Influence matrix for 12 tasks.  (b) Influence distribution for 12 tasks.

Figure 3: This figure provides a visual representation of the mutual influence between tasks. The left-hand side displays the average influence matrix, as defined in Eq. 7, which reveals significant mutual influences existing among the COPs of the same type. Meanwhile, the right-hand side illustrates the influence value, as defined in Eqs. 3, 4, throughout the training process, further demonstrating the extensive mutual impacts among the COPs of the same type and the less pronounced interactions between COPs of different types.

influence matrix computed using Eq. 7 possesses a diagonal-like block structure. This phenomenon suggests a strong correlation between the same type of COP with different problem scales, which is not present within different types of COPs due to the corresponding elements being insignificant. Furthermore, within the same type of COP, we observe that the effect of training a task on other tasks lessens with the increase in the difference of problem scales. Hence, training combinarotial neural solvers on one problem scale leads to higher benefits on similar problem scales than on those that are further away. For instance, the influence of training TSP-20 on TSP-50 is 0.1007, which is higher than the influence on TSP-100, which is $-0.1196$. Similarly, training TSP-100 on TSP-50 has a larger influence than that on TSP-20, as can be observed from influences of $-0.0354$ and $-0.0978$, respectively; **(2)** Figure 3b presents a visualization of the influence resulting from Eq. 3, 4 over the course of the training process. Each point in the chart represents the influence of a particular task on another task at a specific time step. Notably, tasks belonging to the same type of COP are highly influential towards each other due to the large variance of their influence values. Conversely, influences between different types of COPs are negligible, evident from the influence values being concentrated around 0. This striking observation showcases that the employed combinatorial neural solver and algorithm, AM (Kool et al., 2019) and POMO (Kwon et al., 2020), segregate the gradient space into distinct orthogonal subspaces, and each of these subspaces corresponds to a particular type of COP. Furthermore, this implies that the gradient of training each variant of COP is situated on a low-dimensional manifold. As a result, we are motivated to develop more parameter-efficient neural solver backbones and algorithms.

## 5 CONCLUSIONS

In the era of large models, training a unified neural solver for multiple combinatorial tasks is in increasing demand, whereas such a training process can be prohibitively expensive. In this paper, given limited training budgets or resources, we propose an efficient training framework to boost the training of unified multi-task combinatorial neural solvers with a multi-armed bandit sampler. To achieve this, we perform the theoretical loss decomposition, resulting in the meaningful influence matrix that can reveal the intrinsic task relations among different COP tasks, providing evidence for several empirical observations in the area of learning to optimize. We believe that this framework can be powerful for multi-task learning in a broader sense, especially in scenarios where resources are limited, and generalization is crucial. It can also help analyze task relations in the absence of priors. Furthermore, the proposed framework is model-agnostic, which makes it applicable to any existing neural solvers. Different neural solvers may produce varying results on the influence matrix, and a perfect neural solver may gain mutual improvements even from different types of COPs. Therefore, there is an urgent need to study the unified backbone and representation method for solving COPs.

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

## A    PROBLEM DESCRIPTION

**Traveling Salesman Problem (TSP)** - The objective is to determine the shortest possible route that visits each location once and returns to the original location. In this study, we limit our consideration to the two-dimensional euclidean case, where the information for each location is presented as $(x_i, y_i) \in \mathbb{R}^2$ sampled from the unit square.

**Vehicle Routing Problem (VRP)** - The Capacitated VRP (CVRP) (Toth & Vigo, 2014) consists of a depot node and several demand nodes. The vehicle begins and ends at the depot node, travels through multiple routes to satisfy all the demand nodes, and the total demand for each route must not exceed the vehicle capacity. The goal of the CVRP is to minimize the total cost of the routes while adhering to all constraints.

**Orienteering Problem (OP)** - The Orienteering Problem (OP) is a variant of the Traveling Salesman Problem (TSP). Instead of visiting all the nodes, the objective is to maximize the total prize of visited nodes within a total distance constraint. Unlike the TSP and the Vehicle Routing Problem (VRP), the OP does not require selecting all nodes.

**Knapsack Problem (KP)** - The Knapsack Problem strives to decide which items with various weights and values to be placed into a knapsack with limited capacity fully. The objective is to attain the maximum total value of the selected items while not surpassing the knapsack's limit.

## B    LOSS DECOMPOSITION

*Proofs of proposition 1:* We consider the loss $L_j^i(\Theta^i)$ for task $T_j^i$ at time $t_2$ based on mean value theorem and take the first order Taylor expansion for $t_1 < t_2$:

$$
\begin{aligned}
L_j^i(\Theta^i(t_2)) = L_j^i(\Theta(t_2)) &= L_j^i(\Theta(t_1)) + \nabla^T L_j^i(\Psi^i(t_1))(\Theta(t_2) - \Theta(t_1)) \\
&= L_j^i(\Theta^i(t_1)) - \nabla^T L_j^i(\Psi^i(t_1)) \sum_{t=t_1}^{t_2} \eta_t \nabla_{\Theta^i} L(\Theta(t))
\end{aligned}
\tag{9}
$$

where $\Psi^i(t_1)$ is some vector lying between $\Theta^i(t_1)$ and $\Theta^i(t_2)$. Suppose task $T_j^i$ is selected for $c_j^i$ times between time step $t_1$ and $t_2$, we then study the term in the case of all tasks are selected at time step $t$:

$$
\begin{aligned}
\nabla_{\Theta^i} L(\Theta(t)) &= (\nabla_{\theta^{\text{share}}}^T L(\Theta(t)), \nabla_{\theta^i}^T L(\Theta(t)))^T \\
&= (\sum_{p=1}^{K} \sum_{q=1}^{n_p} \nabla_{\theta^{\text{share}}}^T L_q^p(\Theta^p(t)), \sum_{j=1}^{n_i} \nabla_{\theta^i}^T L_j^i(\Theta(t)))^T \\
&= \underbrace{(\nabla_{\theta^{\text{share}}}^T L_j^i(\Theta^i(t)), \nabla_{\theta^i}^T L_j^i(\Theta(t)))^T}_{(a)\text{ gradients of training task } T_j^i} + \underbrace{(\sum_{\substack{q=1 \\ q \neq j}}^{n_i} \nabla_{\theta^{\text{share}}}^T L_q^i(\Theta^i(t)), \sum_{\substack{q=1 \\ q \neq j}}^{n_i} \nabla_{\theta^i}^T L_q^i(\Theta(t)))^T}_{(b)\text{ gradients of training task } \{T_q^i, q \neq i\}} \\
&\quad + \underbrace{(\sum_{\substack{p=1 \\ p \neq i}}^{K} \sum_{q=1}^{n_p} \nabla_{\theta^{\text{share}}}^T L_q^p(\Theta^p(t)), \mathbf{0})^T}_{(c)\text{ gradients of training task } \{T_q^p, p \neq i\}} \\
&= \nabla L_j^i(\Theta^i(t)) + \sum_{\substack{q=1 \\ q \neq j}}^{n_i} \nabla L_q^i(\Theta^i(t)) + (\sum_{\substack{p=1 \\ p \neq i}}^{K} \sum_{q=1}^{n_p} \nabla_{\theta^{\text{share}}}^T L_q^p(\Theta^p(t)), \mathbf{0})^T.
\end{aligned}
\tag{10}
$$

The terms $(a), (b)$ and $(c)$ in Eq. 10 mean the gradients leading by the training of task $T_j^i$, the same kind of COP $\{T_q^i, q \neq j\}$ and other kinds of COPs $\{T^p, p \neq i\}$, respectively. After combining Eq. 9

and 10, we obtain

$$L_j^i(\Theta^i(t_2)) - L_j^i(\Theta^i(t_1))$$

$$= - \underbrace{(\nabla^T L_j^i(\Psi^i(t_1)) \sum_{t=t_1}^{t_2} \mathbb{1}(a_t = T_j^i) \eta_t \nabla L_j^i(\Theta^i(t))}_{(a)\ \text{effects of training task } T_j^i:\ e_j^i(t_1 \rightarrow t_2)} + \underbrace{\nabla^T L_j^i(\Psi^i(t_1)) \sum_{\substack{q=1 \\ q \neq j}}^{n_i} \sum_{t=t_1}^{t_2} \mathbb{1}(a_t = T_q^i) \eta_t \nabla L_q^i(\Theta^i(t))}_{(b)\ \text{effects of training task } \{T_q^i, q \neq j\}:\ \{e_q^i((t_1 \rightarrow t_2)), q \neq j\}}$$

$$+ \underbrace{\nabla_{\theta^{\text{share}}}^T L_j^i(\Psi^i(t_1)) \sum_{\substack{p=1 \\ p \neq i}}^{K} \sum_{q=1}^{n_p} \sum_{t=t_1}^{t_2} \mathbb{1}(a_t = T_q^p) \eta_t \nabla_{\theta^{\text{share}}}^T L_q^p(\Theta^p(t)))}_{(c)\ \text{effects of training task } \{T_q^p, p \neq i\}: \{e_q^p(t_1 \rightarrow t_2), q=1,2,...,n_p, p \neq i\}},$$

$$(11)$$

where $\mathbb{1}(a_t = T_j^i)$ is the indicator function which is introduced here because we only select one task at each time step, taking 1 if selecting task $T_j^i$ at time step $t$, 0 otherwise. $\qquad \square$

Adam optimizer (Kingma & Ba, 2015) is more widely used and popular in practice than standard gradient descent . Accordingly, we derive the loss decomposition for Adam optimizer in a manner consistent with the previous method. We first summarize the update rule of Adam as follows:

$$\Theta(t) = \Theta(t-1) + \alpha \frac{\sqrt{\sum_{i=1}^{t-1} \beta_2^{t-i}}}{\sum_{i=1}^{t-1} \beta_1^{t-i}} \frac{\sum_{i=1}^{t} \beta_1^{t-i} g_i}{\sqrt{\sum_{i=1}^{t} \beta_2^{t-i} ||g_i||^2} + \epsilon}$$

$$= \Theta(t-1) + \eta_t \sum_{i=1}^{t} \beta_1^{t-i} g_i$$

where $g_i = \nabla J(\Theta_{i-1})$ and $g_0 = \mathbf{0}, \eta_t = \frac{\sqrt{\sum_{i=1}^{t-1} \beta_2^{t-i}}}{\sum_{i=1}^{t-1} \beta_1^{t-i}} \frac{1}{\sqrt{\sum_{i=1}^{t} \beta_2^{t-i} ||g_i||^2} + \epsilon}, \eta_i, i = 1, 2$ are exponential average parameters for the first and second order gradients. Our assumption is that sharing the second moment term correction for all tasks can be easily implemented by using a single optimizer during training.

Given that the update is predicated on the optimization trajectory's history, we can use comparable calculations in gradient descent to infer Adam's contribution breakdown. Starting at the same point:

$$L_j^i(\Theta^i(t_2)) = L_j^i(\Theta^i(t_1)) + \nabla^T L_j^i(\Psi^i(t_1))(\Theta^i(t_2) - \Theta^i(t_1))$$

$$= L_j^i(\Theta^i(t_1)) - \nabla^T L_j^i(\Psi^i(t_1)) \sum_{t=t_1}^{t_2} \eta_t \sum_{k=1}^{t} \beta_1^{t-k} \nabla L(\Theta^i(k-1)),$$

then taking Eq. 10 into $\nabla L(\Theta^i(k-1))$, we have

$$L_j^i(\Theta^i(t_2)) - L_j^i(\Theta^i(t_1))$$

$$= - \underbrace{(\nabla^T L_j^i(\Psi^i(t_1)) \sum_{t=t_1}^{t_2} \mathbb{1}(a_t = T_j^i) \eta_t \sum_{k=1}^{t} \beta_1^{t-k} \nabla L_j^i(\Theta^i(k-1))}_{(a)\ \text{effects of training task } T_j^i:\ e_j^i(t_1 \rightarrow t_2)}$$

$$+ \underbrace{\nabla^T L_j^i(\Psi^i(t_1)) \sum_{\substack{q=1 \\ q \neq j}}^{n_i} \sum_{t=t_1}^{t_2} \mathbb{1}(a_t = T_q^i) \eta_t \sum_{k=1}^{t} \beta_1^{t-k} \nabla L_q^i(\Theta^i(k-1))}_{(b)\ \text{effects of training task } \{T_q^i, q \neq j\}:\ \{e_q^i((t_1 \rightarrow t_2)), q \neq j\}}$$

$$(12)$$

$$+ \underbrace{\nabla_{\theta^{\text{share}}}^T L_j^i(\Psi^i(t_1)) \sum_{\substack{p=1 \\ p \neq i}}^{K} \sum_{q=1}^{n_p} \sum_{t=t_1}^{t_2} \mathbb{1}(a_t = T_q^p) \eta_t \sum_{k=1}^{t} \beta_1^{t-k} \nabla_{\theta^{\text{share}}} L_q^p(\Theta^p(k-1)))}_{(c)\ \text{effects of training task } \{T_q^p, p \neq i\}: \{e_q^p(t_1 \rightarrow t_2), q=1,2,...,n_p, p \neq i\}},$$

Three similar parts are obtained finally.

## C  DISCUSSION ON THE BANDIT ALGORITHM AND UPDATE FREQUENCY

As shown in Equation 2, the effect of the training task $T_q^p$ on $T_j^i$ can be computed as

$$\nabla^T L_j^i(\Psi^i(t_1)) \mathbb{1}(a_t = T_q^p) \cdot \nabla L_q^p(\Theta^p(t)).$$

This is subject to the indicator function $\mathbb{1}(a_t = T_q^p)$, which determines whether the task $T_q^p$ is selected at time step $t$. In this bandit setting, there are several observations to note: **(1)** If $t_2 - t_1 = 1$, only one selected task exists, and thus only one column vector in $M(t_1 \rightarrow t_2)$ can be non-zero; **(2)** If task $T_q^p$ is not selected for training, the gradient information $\nabla^T L_q^p(\Theta^p(t))$ cannot be obtained. Further details regarding gradient approximation configuration can be found in Appendix E. **(3)** Stochastic gradient-based methods are commonly used to optimize parameters. However, frequent updates can lead to inaccurate or even incorrect influence estimation (refer to Fig. 5a for the loss during training). Based on these observations, the following tips are highlighted: **(1)** For the stability and accuracy of the gradients, it is recommended to involve more than one step in the process of collecting gradient information. However, having an overly slow update frequency may yield incorrect results due to the lazy update of the bandit algorithm; **(2)** When the update frequency is larger than 1, UCB family algorithms are unsuitable as they tend to greedily select the same task in the absence of updates. Therefore, the update frequency is a crucial hyper-parameter to specify, and Thompson Sampling and adversary bandit algorithms are suitable in this framework due to their higher level of randomness.

Based on above discussions, we present empirical evidence and elaborate on the details. We performed experiments for the 12 tasks under small budgets, with five repetitions each. Five update frequencies were considered: 6, 12, 24, 36, and 48. The performances w.r.t. optimality gap are presented in Figure 4 and furthere results are in Appendix H.

**Effects of bandit algorithms.** The four algorithms considered are: Exp3, Thompson Sampling (TS), Exp3R, and Discounted Thompson Sampling (DTS). They have more exploration characteristics than UCB family algorithms with update delays. Moreover, Exp3R and DTS have the capability to handle changing environments. According to Fig. 4, TS performs the worst among these four algorithms, as it fails to handle potential adversaries and changing environments. DTS performs more robustly than TS since it involves a discounted factor. Exp3 and

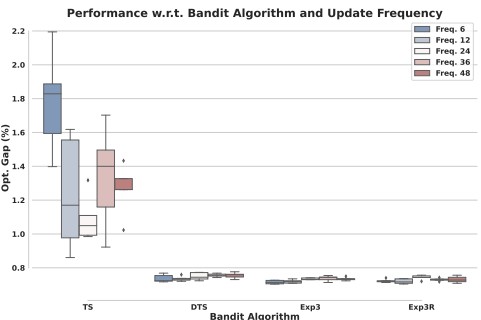

Figure 4: This figure presents the comparison results of various bandit algorithms and update frequencies in terms of optimality gap (%).

Exp3R provide good results because they are able to handle adversaries and detect environmental changes. However, Exp3R does not perform significantly better than Exp3 due to the neural solver's gradual and slow improvement, resulting no abrupt changes for Exp3R to detect. Based on the observed performance, it appears that simple procedures such as introducing a discounted factor in DTS and basic adversary bandit algorithms such as Exp3 are sufficient for handling our case.

**Effects of update frequency.** The update frequency affects the accuracy of influence information approximation and the tension in the bandit algorithm. Appropriate selections must balance these two factors. Figure 4 shows that the frequency of 12 generally yields the best results across different bandit algorithms. DTS and Exp3 exhibit deteriorating performance with higher frequencies, resulting from numerous lazy updates. By contrast, Exp3R does not have this property because increasing the frequency helps detect changing points more quickly. As a consequence, the number of tasks (12 in our case) appears to be an appropriate empirical choice to balance these two factors.

## D  COMPUTATION COMPLEXITY OF MTL METHODS

In this part, we will make a detailed comparison on the computation complexity between our method and other typical MTL methods. We first define some notations for the time complexity:

Table 5: Computation complexity of basic operators.

| Task Num | Dim. Param. | Complexity of FF | Complexity of BP |
|---|---|---|---|
| $N$ | $D_i$ | $F_i$ | $B_i$ |

where $D_i$, $F_i$ and $B_i$ are the dimension of parameters, the computation cost of feed-forward and backward for task $i$, and we denote $D = \max\{D_i, i = 1, 2, ..., N\}, F = \max\{F_i, i = 1, 2, ..., N\}, B = \max\{B_i, i = 1, 2, ..., N\}$. We analyse the computation complexity for Bandit-MTL, PCGrad, Nash-MTL, Uncertainty-Weighting (UW) and our method, results are shown as follows:

Table 6: Computation complexity of different MTL methods for one training time slot. "Basic" measures the computation for the feed-forward and backward process, "Extra" measures the extra computations used for guiding MTL, and "All" is the sum of them.

| | Naive-MTL | Bandit-MTL | PCGrad | Nash-MTL | UW | IMTL | CAGrad | cOurs |
|---|---|---|---|---|---|---|---|---|
| Basic | $\mathcal{O}\left(N(F+B)\right)$ | $\mathcal{O}\left(N(F+B)\right)$ | $\mathcal{O}\left(N(F+B)\right)$ | $\mathcal{O}\left(N(F+B)\right)$ | $\mathcal{O}\left(N(F+B)\right)$ | $\mathcal{O}\left(N(F+B)\right)$ | $\mathcal{O}\left(N(F+B)\right)$ | $\mathcal{O}\left(F+B\right)$ |
| Extra | 0 | $\mathcal{O}(N)$ | $\mathcal{O}(ND)$ | $\mathcal{O}(N^2D)$ | $\mathcal{O}(1)$ | $\mathcal{O}(ND+N^3)$ | - | $\mathcal{O}(ND)$ |
| All | $\mathcal{O}\left(N(F+B)\right)$ | $\mathcal{O}\left(N(F+B+1)\right)$ | $\mathcal{O}\left(N(F+B+D)\right)$ | $\mathcal{O}\left(N(F+B+ND)\right)$ | $\mathcal{O}\left(N(F+B)\right)$ | $\mathcal{O}\left(N(F+B+N^2D)\right)$ | - | $\mathcal{O}(F+B+ND)$ |

where "Basic" measures the computation for the feed-forward and backward process, "Extra" measures the extra computations used for guiding MTL, and "All" is the sum of them.

We ignore the complexity of sampling from a discrete distribution with $N$ elements, e.g. sampling an arm in MAB algorithm. What's more, we also ignore the optimization process in Nash-MTL and UW because they are quite efficient to compute. From the results in the Table 6, our method has moderate extra computation costs comparing with other methods, however, when considering the overall computation cost, our method achieves the lowest complexity because we only need to perform one feedforward-backward process which is the most time-consuming part during training.

## E  EXPERIMENTAL SETTINGS

**Model structure** - We adopt the same model structures as in POMO (Kwon et al., 2020) to build our model. To train various COPs in a unified model, we use a separate MLP on top of the model for each problem, which we call *Header*. This header facilitates correlation of input features with different dimensions. For TSP, we use two-dimensional coordinates, $\{(x_i, y_i), i = 1, 2, ..., n\}$, as input, while CVRP and OP have additional constraints on customer demand and vehicle capacity, in addition to two-dimensional coordinates. Hence, their input dimensions are 3 and 3, respectively. Moreover, in OP, the prize is assigned based on the distance between the node and the depot node, following the setting in AM (Kool et al., 2019). The KP takes two-dimensional inputs, $\{(w_i, v_i), i = 1, 2, ..., n\}$, with $w_i$ and $v_i$ representing the weight and value of each item, respectively. As such, we introduce four kinds of *Header* to embed features with different dimensions to 128. The embeddings obtained from the *Header* are then passed through a shared *Encoder*, composed of six encoder layers based on the Transformer (Vaswani et al., 2017). Finally, we employ four type-specific *Decoder*s, one for each COP, to make decisions in a sequential manner. The shared *Encoder* has the bulk of the model's capacity because the *Header* and *Decoder* are lightweight 1-layer MLPs. Furthermore, when solving a specific COP, we only need to use the relevant *Encoder*, *Header*, and *Decoder* for evaluation. Since the model size is precisely the same, the inference time required is similar to that of single-task learning.

**Hyperparameters** - In each epoch, we process a total of 100×1000 instances with a batch size of 512. The POMO size is equal to the problem scale, except for KP-200, where it is 100. We optimize the model using Adam (Kingma & Ba, 2015) with a learning rate of 1e-4 and weight decay of 1e-6. The training of the model involves 1000 epochs in the standard setting. The learning rate is decreased by 1e-1 at the 900th epoch. During the first epoch, we use the bandit algorithm to explore at the beginning of the training process. We then collect gradient information by updating the bandit algorithm with every 12 batches of data. The model is trained using 8 Nvidia Tesla A100 GPUs in parallel, and the evaluations are done on a single NVIDIA GeForce RTX 3090.

**Approximation of gradients** - Another issue is the approximation of $\nabla L_q^i(\Theta^i(t))$ in Eq. 3 and $\nabla_{\theta^{\text{share}}} L_q^p(\Theta^p(t))$ in Eq. 4 when tasks $T_q^i$ and $T_q^p$ are not selected during the update interval. To obtain an approximation, we use the most recent gradient information collected from the last time they were selected to train. This approximation is necessary because training task $T_j^i$ can change the values of $\Theta^i$ and $\theta^{\text{share}}$, which can affect other training tasks. Considering all these changes is necessary to accurately measure the influences of training $T_j^i$ on other tasks.

**Bandit settings** - We utilized the open-source repository (Besson, 2018) for implementing the bandit algorithms in this study with default settings.

## F   LOSS AND GRADIENT NORM OF EACH TASK

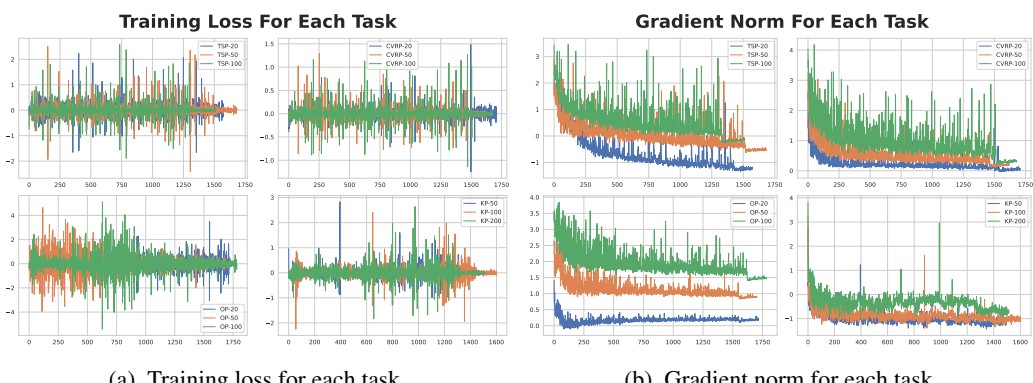

(a) Training loss for each task.     (b) Gradient norm for each task.

Figure 5: Training loss and gradient norm for each task in the log-scale.

One intuitive method of measuring the effect of training is to calculate the ratio of losses between adjacent training sessions. These ratios can be used to calculate training rewards for each corresponding task. However, as shown in Figure 5a, this method of calculating rewards is not effective because they are not sufficiently distinct to guide the training process properly.

Computing the inner products of corresponding gradients to analyze how training one task affects the others can lead to a misleading calculation of rewards and training process. Figure 5b visualizes gradient norms for each task in the logarithmic scale. We observe that the gradient norms are not in the same scale, which becomes problematic when jointly training different COP types. In such cases, the rewards of certain COP types (such as CVRP in our experiments) may dominate the rewards of other types.

## G   DEMONSTRATION OF THE BANDIT ALGORITHMS

This section presents detailed information on various bandit algorithms, as shown in Fig. 6, including the selection count and average return for each task. It is evident that TS algorithm dominates in all 12 tasks, leading to poor performance on tasks where training is limited. In contrast, other bandit algorithms maintain balance across all tasks, resulting in better average results.

## H   FURTHER RESULTS ON THE BANDIT ALGORITHM SELECTION AND UPDATE FREQUENCY

In Appendix C, we examine the impact of bandit algorithms and update frequency on 12 tasks, specifically on the average optimality gap. We also analyze the effect of these two factors on the influence matrix, which is presented in this section. For ease of understanding, a visual aid is included in Figure 7. By combining the results from Figure 3 and Figure 7, we can infer that influence matrices derived from DTS, Exp3, and Exp3R with an update frequency of 6 and 12 comply with the rule specified in Section 4.2. However, the TS algorithm disregards this rule due to its inability to handle

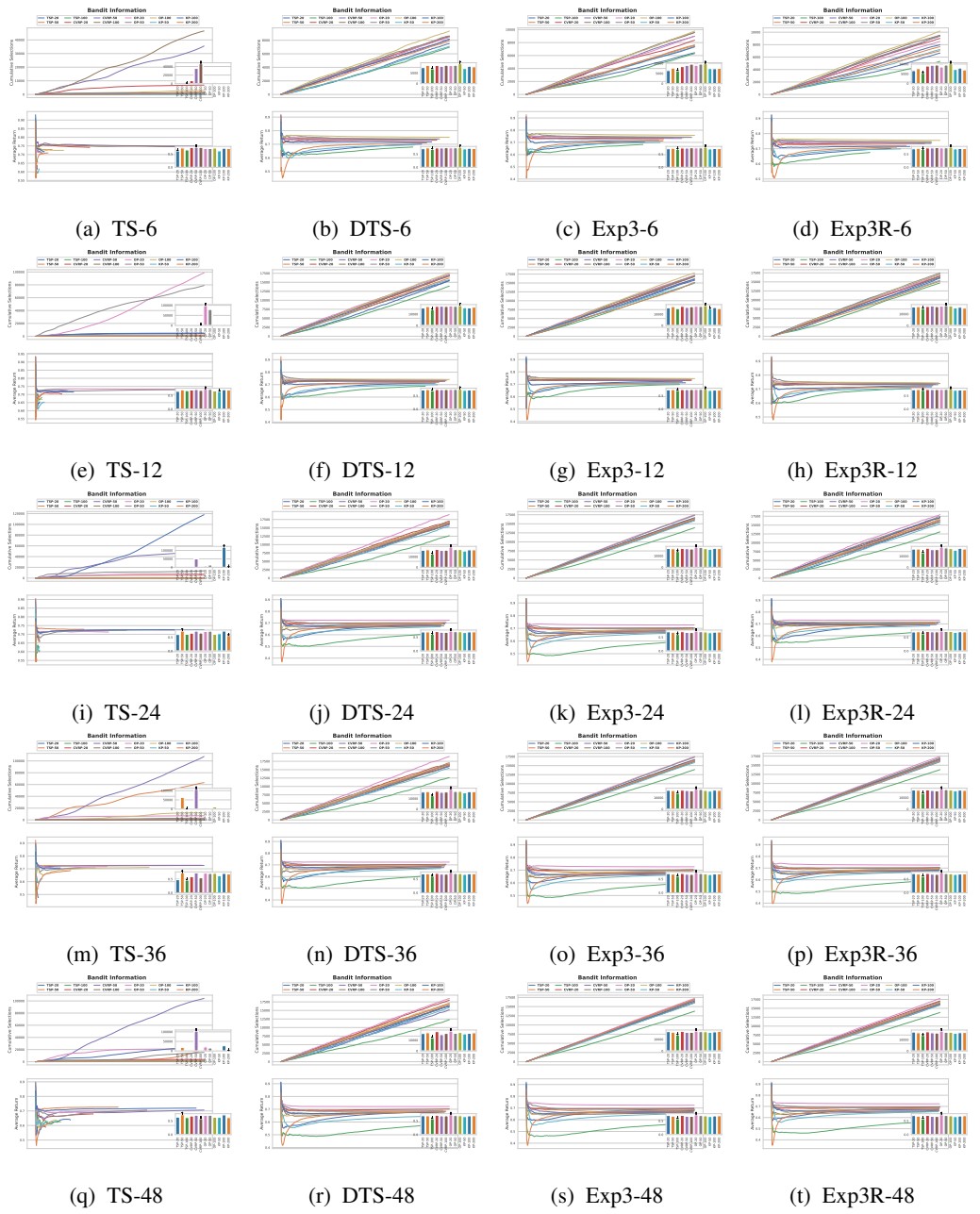

(a) TS-6  (b) DTS-6  (c) Exp3-6  (d) Exp3R-6

(e) TS-12  (f) DTS-12  (g) Exp3-12  (h) Exp3R-12

(i) TS-24  (j) DTS-24  (k) Exp3-24  (l) Exp3R-24

(m) TS-36  (n) DTS-36  (o) Exp3-36  (p) Exp3R-36

(q) TS-48  (r) DTS-48  (s) Exp3-48  (t) Exp3R-48

Figure 6: Further results of the bandit information. The caption of each subfigure "A-B" means the influence matrix obtained by algorithm A with update frequency B.

adversaries and changing environments. Moreover, when the update frequency is increased, the approximation of the influence matrix is impaired due to the lazy update of bandit algorithms. As a result, utilizing the number of tasks as the update frequency appears to be a sound decision, as it not only improves performance but also enhances interpretability.

# I    ADDITIONAL EXPERIMENTS ON OTHER DOMAINS

We select the challenge domain on Time Series to evaluate the performance of our method. Following the common practice in this domain, there are multiple series in one piece of data and the prediction on each series is seen as a task. [3]We consider Long-term Forecasting tasks comprising ETT (4 subsets), Weather, Exchange and ILI datasets, and Imputation task comprising ETT and Weather. The backbone is AutoFormer (Wu et al., 2021a) and all the experimental settings keep the same as the original paper.

Table 7: For Long-term Forecasting tasks, all the results are averaged from 4 different prediction lengths, that is $\{24, 36, 48, 60\}$ for ILI and $\{96, 192, 336, 720\}$ for the others. "Baseline" provides the the results in the original paper. Results in bold mean achieving the best performance among all methods.

| Method | ETT-h1 MSE | ETT-h1 MAE | ETT-h2 MSE | ETT-h2 MAE | ETT-m1 MSE | ETT-m1 MAE | ETT-m2 MSE | ETT-m2 MAE | Whether MSE | Whether MAE | Exchange MSE | Exchange MAE | ILI MSE | ILI MAE |
|---|---|---|---|---|---|---|---|---|---|---|---|---|---|---|
| MTL | 0.496 | 0.487 | 0.450 | 0.459 | 0.588 | 0.517 | 0.327 | 0.371 | 0.338 | 0.382 | 0.613 | 0.539 | 3.006 | **1.161** |
| Bandit-MTL | 0.438 | 0.420 | 0.398 | 0.363 | 0.533 | 0.643 | 0.304 | 0.220 | 0.327 | 0.254 | 0.319 | 0.181 | 1.424 | 3.955 |
| UW | 0.420 | 0.385 | 0.400 | 0.359 | 0.502 | 0.557 | 0.325 | 0.236 | **0.308** | **0.231** | 0.287 | 0.153 | 1.425 | 3.942 |
| CAGrad | 0.468 | 0.466 | 0.390 | 0.351 | 0.477 | 0.488 | 0.304 | 0.220 | 0.312 | 0.241 | 0.310 | 0.174 | 1.411 | 3.856 |
| IMTL-G | 0.445 | 0.423 | 0.392 | 0.352 | **0.465** | **0.462** | 0.303 | 0.219 | 0.319 | 0.245 | **0.286** | **0.153** | 1.399 | 3.776 |
| Nash-MTL | 0.468 | 0.472 | 0.409 | 0.370 | 0.468 | 0.483 | 0.310 | 0.225 | 0.315 | 0.240 | 0.302 | 0.165 | **1.376** | 3.806 |
| Ours | **0.418** | **0.385** | **0.383** | **0.343** | 0.506 | 0.547 | **0.299** | **0.215** | 0.360 | 0.277 | 0.333 | 0.193 | 1.689 | 5.189 |

Results show that there are no consisting best methods for all datasets, however, our method can achieve the best performance consistently on 3 out of 7 datasets.

Table 8: For Imputation tasks, time series are randomly masked $\{12.5\%, 25\%, 37.5\%, 50\%\}$ time points in length-96. The results are averaged from 4 different mask ratios. Results with underline mean achieving the best performance among all methods and those in bold mean achieving the best among all MTL methods.

| Method | ETT-h1 MSE | ETT-h1 MAE | ETT-h2 MSE | ETT-h2 MAE | ETT-m1 MSE | ETT-m1 MAE | ETT-m2 MSE | ETT-m2 MAE | Weather MSE | Weather MAE |
|---|---|---|---|---|---|---|---|---|---|---|
| Baseline | 0.103 | 0.214 | 0.055 | 0.156 | 0.051 | 0.150 | 0.029 | 0.105 | 0.031 | 0.057 |
| Bandit-MTL | 0.324 | 0.201 | 0.437 | 0.414 | 0.579 | 0.576 | **0.603** | 0.792 | **0.255** | **0.154** |
| UW | **0.268** | **0.143** | 0.354 | 0.280 | 0.669 | 0.767 | 0.774 | 1.124 | 0.391 | 0.324 |
| CAGrad | 0.269 | 0.144 | 0.354 | 0.267 | 0.593 | 0.605 | 0.640 | 0.747 | 0.347 | 0.257 |
| IMTL | 0.270 | 0.145 | 0.353 | 0.266 | 0.594 | 0.605 | 0.681 | 0.854 | 0.388 | 0.337 |
| Nash-MTL | 0.268 | 0.143 | **0.347** | **0.255** | 0.637 | 0.694 | 0.693 | 0.866 | 0.489 | 0.494 |
| Ours | 0.300 | 0.174 | 0.411 | 0.366 | **0.473** | **0.384** | 0.609 | **0.734** | 0.283 | 0.174 |

From these results, our method performs well in some cases, but generally speaking, there is no one universal approach which can handle all tasks or even on all datasets in a task .

# J    STABILITY OF EACH METHOD

We demonstrate the stability of each model obtained by the specific methods on 10000 test instances from each COP and the corresponding error bar plot is shown in Fig. 8. It is generally accepted that longer training epochs lead to reduced standard variance for each method. Additionally, our method

---

[3]For forecasting and imputation tasks, we ignore the Electricity and Traffic dataset because all MTL methods meet out of memory errors because there are too many tasks.

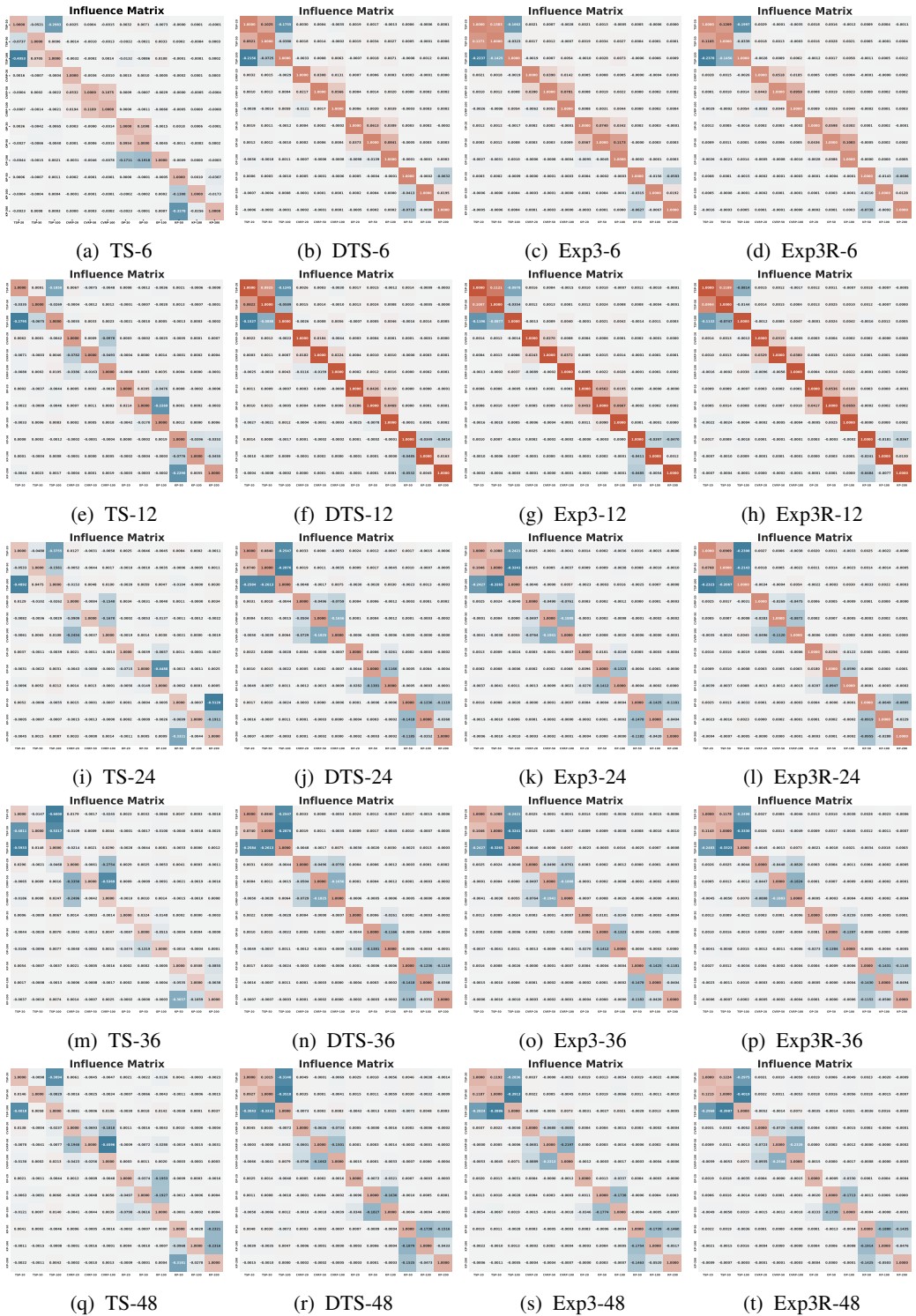

Figure 7: Further results of the influence matrix on the selection of bandit algorithms and update frequency. The caption of each subfigure "A-B" means the influence matrix obtained by algorithm A with update frequency B.

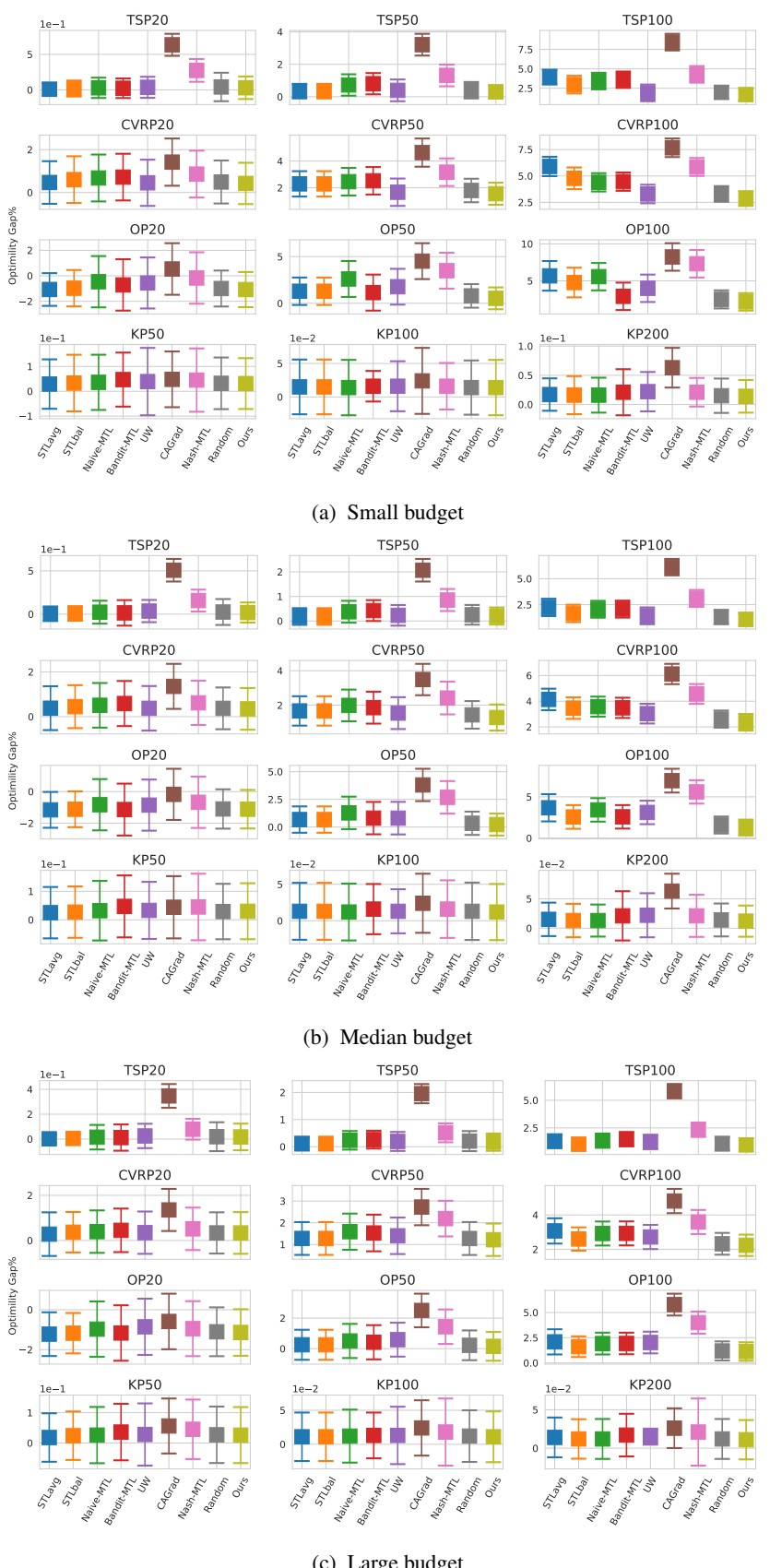

Figure 8: Stability of the model obtained by different methods on 10000 instances from each COP with different budget allocations.

produces a model with the most stable performance in most scenarios when compared to other MTL methods across almost all cases.

