# OpenReview forum: "Efficient Training of Multi-task Combinarotial Neural Solver with Multi-armed Bandits"
_ICLR.cc/2024/Conference — Submitted to ICLR 2024_

### Official Review · Reviewer_7uDV · 2023-10-23

**Soundness:** 3 good
**Presentation:** 3 good
**Contribution:** 3 good
**Rating:** 5
**Confidence:** 4

**Summary:**

The paper studies the problem of training a neural solver for multi-task combinatorial optimization. It proposes to use a bandit algorithm to choose the task instances for training to optimize the final model performance over all tasks. It proposes to use the average change of the training loss (approximate by first order gradient signals) due to the chosen task as the reward. Experimental studies on 12 tasks of various size and type are done using EXP3 algorithm and compared to single task learning and other multi-task learning algorithms.

**Strengths:**

1. Significance: The paper formulates an interesting and useful problem. The proposed algorithm outperforms many existing algorithms.
2. Clarity and quality: the paper is well written and flows smoothly. The problem is clearly motivated and an effective solution is proposed.

**Weaknesses:**

1. Novelty: given the specific structure of the problem, the paper could improvise more specifically tailored bandit algorithms for the problem; a simple example would be using a structured/contextualized bandit algorithm that takes into account the complexity and type of the tasks.

2. Section 3.2 could benefit from some clarifications: the reward design could be simplified and avoid some repetition of notations. On the other hand, the discussion around Assumption 1 needs more elaboration; it is not clear how the equality is established or if it is an approximation indeed.

**Questions:**

1. $\eta_t$ is never defined.

2. Given that neural solvers usually have a large number of parameters, what do you think about the space complexity of saving the gradient information in step 2 of Algorithm 1?

3. In Section 4.1, it looks like the STL model trained on the large instances perform on par with your model, based on the "Mean Results on *<COP>*" plots, except only for CVRP. Then why do you think "Our method can handle various types of COPs under the same number of training epochs, which is **impossible for STL** ... "?

---

> ### Author Response · Authors · 2023-11-18
>
> Thank you for the confirmations and suggestions.
> # Response to the weaknesses
> 1. The proposed method in this work is agnostic to any bandit algorithms. While it is true that employing more sophisticated bandit algorithms could enhance performance by leveraging pertinent prior knowledge, it's common to lack such priors. Therefore, we focus on the most general case, wherein the environment within the bandit context dynamically changes due to training neural solvers. We choose Exp3 to address potential adversarial effects under these circumstances.
> 2. In the revised version of Section 3.2, we have made efforts to eliminate redundant content for improved clarity.
>
>     Assumption 1 comes from previous impactful works so we consider it to be a reasonable starting point. From this assumption, the similarity in this work is defined as: $\operatorname{Similarity}\left(T_j^i, T_j^i\right):=\cos \left(\nabla L_j^i\left(\Psi^i\left(t_1\right)\right), \sum_{t=t_1}^{t_2} \eta_t\mathbb{1}\left(a_t=T_j^i\right) \nabla L_j^i\left(\Theta^i(t)\right)\right)=1.$ This similarity holds true only when the two gradient vectors share the same direction, i.e., when:$\nabla L_j^i\left(\Psi^i\left(t_1\right)\right)= c\sum_{t=t_1}^{t_2} \eta_t\mathbb{1}\left(a_t=T_j^i\right) \nabla L_j^i\left(\Theta^i(t)\right)$ for some constant $c>0$.  The reward for updating bandits is defined as the cosine similarity, and the impact of $c$ is nullified.
> $$
> \begin{aligned}
> &\operatorname{Similarity}\left(T_j^i, T_j^i\right):=\cos \left(\nabla L_j^i\left(\Psi^i\left(t_1\right)\right), \sum_{t=t_1}^{t_2} \eta_t\mathbb{1}\left(a_t=T_j^i\right) \nabla L_j^i\left(\Theta^i(t)\right)\right)\\\\
> &=\frac{\langle c\sum_{t=t_1}^{t_2} \eta_t\mathbb{1}\left(a_t=T_j^i\right) \nabla L_j^i\left(\Theta^i(t)\right), \sum_{t=t_1}^{t_2} \eta_t\mathbb{1}\left(a_t=T_j^i\right) \nabla L_j^i\left(\Theta^i(t)\right)\rangle}{|| c\sum_{t=t_1}^{t_2} \eta_t\mathbb{1}\left(a_t=T_j^i\right) \nabla L_j^i\left(\Theta^i(t)\right)||· || \sum_{t=t_1}^{t_2} \eta_t\mathbb{1}\left(a_t=T_j^i\right) \nabla L_j^i\left(\Theta^i(t)\right)||}\\\\
> &=\frac{\langle\sum_{t=t_1}^{t_2} \eta_t\mathbb{1}\left(a_t=T_j^i\right) \nabla L_j^i\left(\Theta^i(t)\right), \sum_{t=t_1}^{t_2} \eta_t\mathbb{1}\left(a_t=T_j^i\right) \nabla L_j^i\left(\Theta^i(t)\right)\rangle}{|| \sum_{t=t_1}^{t_2} \eta_t\mathbb{1}\left(a_t=T_j^i\right) \nabla L_j^i\left(\Theta^i(t)\right)||· || \sum_{t=t_1}^{t_2} \eta_t\mathbb{1}\left(a_t=T_j^i\right) \nabla L_j^i\left(\Theta^i(t)\right)||}=1\\\\
> \end{aligned}
> $$
>
> where $\langle\cdot,\cdot\rangle$ is the inner product for vectors. Therefore, setting $c=1$ results in Eq. (8) in the revised paper:$
> \nabla L_j^i\left(\Psi^i\left(t_1\right)\right)= \sum_{t=t_1}^{t_2}\eta_t \mathbb{1}\left(a_t=T_j^i\right) \nabla L_j^i\left(\Theta^i(t)\right).$
> # Response to the questions
> 1. We appreciate the reviewer's observation. $\eta_t$ represents the learning rate used in training neural networks. To enhance clarity, we have included a definition for $\eta_t$ in the revised Proposition 1.
> 2. The computation involves the cosine similarity between all pairs of tasks, resulting in a space complexity of $n N_{\text{share}} + N_{\text{task-specific}}$, where $n$ is the number of tasks, $N_{\text{share}}$ is the parameter count of shared modules, and $N_{\text{task-specific}}$ is the parameter count of task-specific modules. Importantly, this complexity is consistent with that of typical MTL baselines examined in our paper.
> 3. The comparison in Fig. 2 might be perceived as unfair due to the difference in training epochs. Our method demonstrates superiority by training a single model for 1000 epochs compared to 12 models trained under "COP-n," each for 1000 epochs (i.e., $1000$ epochs vs $12\times 1000$ epochs in total). Despite this discrepancy, our neural solver's performance on a specific COP is either better or competitive with the results from 3 models obtained by STL (trained for a total of 3000 epochs).
>
>     The statement "...impossible for STL..." is grounded in the fact that STL exclusively focuses on one type of COP with a specific problem scale. Consequently, STL not only exhibits poor performance on different problem scales due to limited generalization ability but also fails to adapt to various COP types, given the presence of task-specific modules.

---

### Official Review · Reviewer_B49D · 2023-10-24

**Soundness:** 2 fair
**Presentation:** 3 good
**Contribution:** 2 fair
**Rating:** 6
**Confidence:** 3

**Summary:**

This paper proposes to train a multi-task combinatorial neural solver under limited budget constraints. The combinatorial optimization problems chosen for their setting are the Travelling Salesman Problem (TSP), the Capacitated Vehicle Routing Problem (CVRP), the Orienteering Problem (OP), and the Knapsack Problem (KP), and each of them with three problem scales that vary from 20 to 200 tasks. The key idea (as with any multi-task learning setting) is to learn the shared parameters across tasks as well as the parameters for the individual task itself. This loss decomposition is captured in section 3.1. I think the key novelty of their paper lies in designing the reward for tasks so that they can use an MAB (Multi-armed bandit) algorithm to first select a task, calculate the loss, use the designed reward to update the MAB algorithm and proceed to the next iteration. The reward is designed using the cosine similarity function in such a way that it facilitates learning between tasks under the same COP and across tasks under different COPS and an influence matrix is constructed with it. Finally, they use the column sum of this influence matrix (which captures the influence of one task on all others) to get an average reward for each task and use it to select the next task. They conduct an empirical evaluation demonstrating two things mainly: (1) Under identical training budgets, their method effectively solves for multiple COPs by learning a shared representation (influence) rather than solving each task individually; (2) Given the same number of training epochs, their method comes up with a neural solver that demonstrates better generalization capability.

**Strengths:**

1) The paper is of low to moderate significance as the key findings from their empirical results are mostly known. Nevertheless, I find it interesting because they use MABs as a sirt if uncertainty quantification on top of a multi-task neural solver which leads to a practical algorithm.
2) They provide a well-justified theoretical decomposition of their loss function for both GD and ADAM settings.
3) They conduct extensive experiments across all four COP settings under different budgets and scale levels. They demonstrate that their neural solver performs better than typical MTL methods and require less budget.

**Weaknesses:**

1) The writing needs more improvement. For example, they never talk about the specific MAB algorithm they use till Appendix G and H (except one line in the experimental setting). It seems they are using Thompson Sampling, or Discounted Thompson Sampling, or Exp algorithms as base MABs. These choices need more justification.
2) The key findings from their empirical evaluations are mostly known: a) Training combinatorial neural solvers on one problem scale leads to higher benefits on similar problem scales than on those that are further away. b) Negative transfer exists among different tasks. c) Easier tasks require less budget.
3) No analysis of budget allocation is done, even though this is fairly well understood in the multi-task setting (both online and offline).

**Questions:**

1) It is not clear to me how the four different COPS can be considered as different tasks, and how the difference is captured. There is no discussion on this. Moreover from Appendix F, it is not clear how the Gradient norm captures the difference between the four different COPS as well as the scales within a COP. Can the authors elaborate on this?
2) You use the column sum of the influence matrix to calculate the average reward for each task and then select the next task based on that. How do you ensure that you conduct a sufficient exploration/exploitation of task? Don't you want to use an exploration bonus (like UCB) over the average reward \bar{r}^i_j?
3) Why do you only use the column-sum of the rewards, and not the row-sum from the influence matrix? Note that the row-sum denotes the influence of all tasks on one task. Intuitively even that should facilitate learning for an individual task.
4) This question is again related to the diversity of tasks. How is your neural solver affected when some tasks are more difficult than others? How does the budget get proportioned then? What happens to the influence matrix then? Can you elaborate on this?
5) Some design choices in the experiment section need more explanation. For example why the balanced allocation is chosen in 1:2:3 ratio from small to large scale?

**Details Of Ethics Concerns:**

Not applicable.

---

> ### Author Response · Authors · 2023-11-18
>
> Thank you for the efforts on proposing these comments and questions.
> # Response to the weaknesses
> 1. The proposed method affords flexibility in the selection of bandit algorithms, with the primary objective of introducing the concept of employing bandits to dynamically guide training. Due to space constraints, we defer an in-depth discussion of bandit algorithms to Appendix C. And we commit to providing enhanced justifications for the chosen algorithms in the revised version.
> 2. (Also response for strength 1)
>     - Regarding point a), while it is known and intuitively correct, our contribution lies in providing empirical justification for these observations in Fig 3. This not only validates the efficacy of our method but also highlights its potential application on tasks where task relationships are unknown.
>     - For point b), while positive or negative transfers does exist among different tasks, our contribution is quantifying this phenomenon by the influence matrix. Additionally, we introduce a novel observation that different types of COPs may not significantly influence each other during training, representing a new contribution to ML4CO community.
>     - Concerning point c), the understanding that easier tasks require less budget is evident. However, the challenge lies in resource allocation when no priors on the tasks are available, and our proposed method is designed to address this practical concern.
> 3. The analysis of budget allocation is detailed in Appendix G, supplemented with additional descriptions. In summary, Discounted Thompson Sampling, Exp3, and Exp3R, designed to handle changing and potentially adversarial environments, exhibit nearly identical budget allocations. In contrast, naive Thompson Sampling fails to adapt, leading to suboptimal performance.
> # Response to the questions
> 1. In the ML4CO community, prior RL-based works have typically trained one model for a specific type of COP with a designated problem scale. Consequently, addressing various types of COPs with different problem scales can be conceptualized as a MTL problem for two reasons: **a)** The existence of task-specific modules that cannot be universally shared across different COP types. **b)**  Variations in the MDP transition steps occur for different problem scales within a COP.
>
>     Concerning Appendix F, as acknowledged, differences among tasks cannot be effectively captured through the loss. This limitation is explicitly discusses in Section 3.1: "*...such measurement is invalid in our context because there are no significant differences among tasks.*" Consequently, we adopt gradient-based methods to distinguish tasks. In the discussion of Fig. 3(b), we illustrate that using the inner product between gradients as reward can be misleading due to varying gradient norms among tasks. To address this, we employ cosine similarity, focusing solely on the angles between gradients.
> 2. The average reward serves as the basis for updating the bandit algorithm, and the exploration-exploitation procedure is entirely governed by the bandit algorithm.
> 3. As you rightly pointed out, the row-sum signifies the influence of all tasks on one specific task, while the column-sum represents the influence of one task on all tasks collectively, which aligns with the motivation of selecting a task that maximally benefits all tasks.
> 4. Our algorithm prefers tasks that provide maximal benefits to all tasks rather than solely focusing on the most challenging task. To provide further insight, we apply the proposed method on three tasks: TSP20, TSP50, and TSP100, with a clear difficulty level follows the order TSP20<TSP50<TSP100. The resulting budget allocation is distributed in the ratio of 0.326, 0.341, and 0.333, respectively, with a larger proportion of resources allocated to TSP50. This allocation strategy is justified both empirically and through the examination of the influence matrix:**a)** In the ML4CO community, it is well-known that training tasks of intermediate difficulty can benefit those on both sides of the difficulty spectrum.  **b)** The influence matrix, shown below, provides support for this allocation strategy: Training TSP50 has a higher similarity contribution of 0.126 to TSP20 and 0.137 to TSP100, compared to the contribution from TSP100 to TSP20 (0.031) and from TSP20 to TSP100 (0.025).
> |  |   |   |
> |-|-|-|
> | 1. | 0.126 | 0.031 |
> | 0.124 | 1. | 0.153 |
> | 0.026 | 0.137 | 1. |
> |  |   |   |
> 5. The 1:2:3 allocation is deemed appropriate for categorizing tasks into easy-median-hard levels. We intend to convey the idea that better resource allocation yields superior results. As evidenced in Table 2, STL_bla outperforms STL_avg, highlighting the significance of proper resource allocation. While alternative allocation ratios could be explored, our focus lies in demonstrating the effects rather than determining the optimal allocation ratio. Thank you for pointing out this, we have revised the descriptions.

---

> > ### Comment · Reviewer_B49D · 2023-11-21
> > **Answer to Rebuttal**
> >
> > Thank you for your rebuttal and answering some of my questions.
> > - I appreciate the author's response that their work provides empirical justification for using neural combinatorial solvers across tasks with unknown correlations between them.
> > - It is slightly strange that cosine similarity (which doesn't consider the magnitude of the gradients) works better than dot product (which does). Do you have any intuition as to why this is true?
> > - The table is not exactly clear to me. Can you please label the rows and columns? Also can you provide some references for your justification that "In the ML4CO community, it is well-known that training tasks of intermediate difficulty can benefit those on both sides of the difficulty spectrum"?

---

> ### Author Response · Authors · 2023-11-21
>
> Thanks for your feedback.
>
> - Using the inner product can lead to misleading updates in bandit algorithms. In Section 3.2 of our paper, we provide a clear explanation for why cosine similarity is preferable over the inner product. Specifically, we state:
>
>     "*... the inner products of gradients from different tasks can significantly differ at scale. This will mislead the bandit’s update seriously since improvements may come from large gradient values even when they are almost orthogonal.*"
>
>     To further substantiate this intuition, we present Figure 5(b) in Appendix F, which visualizes the gradient norms for each task on a logarithmic scale. The figure clearly demonstrates that the gradient norms are not uniformly distributed on the same scale. Consequently, this discrepancy in scales leads to a preference for training on tasks with larger gradient magnitudes.
>
> - Apologies for the lack of clarity in the table. Here is the revised table with labeled rows and columns:
>
>     |        | TSP20 | TSP50 | TSP100 |
>     |--------|-------|-------|--------|
>     | TSP20  | 1.    | 0.126 | 0.031  |
>     | TSP50  | 0.124 | 1.    | 0.153  |
>     | TSP100 | 0.026 | 0.137 | 1.     |
>
>     Regarding the reference,  we can provide two influential works, namely Figure 3 in [1] and Figure 5 in [2], which provide insights supporting this claim of "*In the ML4CO community, it is well-known that training tasks of intermediate difficulty can benefit those on both sides of the difficulty spectrum.* Both studies demonstrate that the model trained on TSP50 exhibits the best average performance on TSP20, TSP50, and TSP100, in contrast to the models trained on TSP20 and TSP100, which perform poorly on tasks outside their training scale too far. Intuitively, this claim is supported by the fact that TSP50 lies between TSP20 and TSP100 in terms of the problem scale, making it easier to generalize performance to both sides. Conversely, the model trained on TSP20(TSP100) performs poorly on TSP100(TSP20) due to the large discrepancy in scale between TSP20 (TSP100) and TSP100 (TSP20). This observation can be verified by the above influence matrix: The rewards for training each task w.r.t. column sum are 1.151 1.264 and 1.185, where training TSP50 has the largest reward, meaning the most positive effects on all tasks.
>
> [1]. Joshi C K, Cappart Q, Rousseau L M, et al. Learning the travelling salesperson problem requires rethinking generalization[J]. arXiv preprint arXiv:2006.07054, 2020.
>
> [2]. Kool W, Van Hoof H, Welling M. Attention, learn to solve routing problems![J]. arXiv preprint arXiv:1803.08475, 2018.

---

### Official Review · Reviewer_ezHK · 2023-10-30

**Soundness:** 3 good
**Presentation:** 2 fair
**Contribution:** 2 fair
**Rating:** 5
**Confidence:** 3

**Summary:**

This paper proposes a multi-armed bandit (MAB) framework to learn a general neural combinatorial optimization solver. The key idea is to use the neural solver loss gradient to construct an influence matrix, which guides the sampling of the next training task. The sampled tasks are used to co-train a global encoder in an encoder-decoder framework. The proposed method was demonstrated on several combinatorial benchmarks. It is compared against other multi-task learning benchmarks and achieves good performance.

**Strengths:**

The paper is clearly written. Experiments are well designed, and generally has promising results.

The proposed MTL strategy builds on the idea of loss difference to construct an influence matrix (Fifty et al., 2021) and extends it to gradient difference. As far as I am aware, this is a new and interesting contribution.

**Weaknesses:**

1. There is a general lack of motivation for the proposed method throughout the paper.
- How is the technique proposed specific to solving COP? As far as I understand, it could be applied to MTL setting. Should it outperform other MTL baselines in a general scenario? If not, why is it performing well on the COP tasks?

- The paper went on to describe a heuristic reward design, but ultimately why is it better? There is neither theoretical guarantee nor complexity analysis, so the method is somewhat unconvincing to me.

2. Regarding significance:
- Experiment results are generally promising, but error bar should be provided to quantify significance.
- It feels strange that the authors brought up (Fifty et al., 2021) in the intro as the inspiration for the proposed method, but did not compare to it. Is there a specific reason for this?

**Questions:**

1. What is responsible for the efficiency of the proposed method? It is about 15 times faster than the closest method (Table 3) despite only changing the sampling scheme. Can the authors provide a breakdown of how other MTL techniques compute the influence matrix?
2. Can we apply a naive co-training scheme (with random sampling of training tasks) to achieve the same effect as the proposed method? I figure it would serve as a good ablation study.
3. How often does assumption 1 hold in practice?
4. Why are some entries in the diagonal blocks negative (whereas many off-diagonal entries are positive). It feels unintuitive that tasks from the same group have weaker influence on one another than tasks from different groups.
5. Can the method be generalized to account for unseen tasks?

---

> ### Author Response · Authors · 2023-11-18
>
> Thanks for your endorsement.
> # Response to the weaknesses
> 1. We aim to clarify our motivation in the following manner:
>     - The task relationships among COP has been observed but lacks both experimental and theoretical validation. Our approach begins by empirically substantiating these observations through the proposed MTL algorithm. Furthermore, at the implementation level, training neural solvers for COPs typically demands more time compared to tasks in classical MTL benchmarks. Traditional MTL methods, involving simultaneous training of all tasks, prove to be time-consuming and memory-intensive. As evidenced by the comparisons in Table 3, our approach, which selectively trains one task at each time slot, demonstrates efficiency and effectiveness. This is particularly notable in terms of training flexibility and cost-effectiveness, as emphasized in the discussions on page 7 (Table 3) and Appendix D.
>
>         We extend our method to other MTL scenarios, as detailed in Appendix I. The results indicate that there is no single MTL method consistently excelling across all scenarios, and in certain cases, they may even underperform compared to a naive MTL approach (refer to Table 7). However, in scenarios where traditional MTL methods show efficacy, our approach maintains competitive performance (refer to Table 6).
>     -  First of all, the heuristic reward design, grounded in the cosine similarity between gradients from different tasks, aligns with widely accepted practices established in influential prior works (Wang et al., 2020; Yu et al., 2020). Secondly, this design choice is default for us and is derived from fundamental loss decomposition (Proposition 1), providing a theoretical foundation. Experimentally, the results obtained with this reward align with observations in the field of ML4CO. Consequently, our reward design stands as both theoretically sound and empirically validated.
> 2. - We have made significant improvements to our presentation by incorporating error bars, showing the performance on 10,000 test instances from each COP, as shown in Figure 8 in Appendix J of the revised version.
>     - The work by Fifty et al. (2021) focuses on MTL centered around task grouping. This approach necessitates an offline pre-training phase for constructing task relationships, followed by training the resulting groupings for evaluation. Conversely, our work and the considered baselines are all online MTL methods, involving training the model on-the-fly and conducting evaluations immediately post-training.
> # Response to the questions
> 1. The enhanced efficiency of our proposed method stems from the changing in the training paradigm. Unlike typical MTL methods that simultaneously train all tasks at each training time slot, our method adopts a more targeted approach by selecting and training only one task per slot. This streamlining significantly reduces computational load and accelerates the training process. Regarding the computation of the influence matrix, it's noteworthy that other MTL methods typically employ specific heuristics to balance the loss from each task, and they do not compute an influence matrix.
> 2. The naive co-training strategy, referred to as Naive-MTL, is presented in Table 2. In addition, we include the results of random sampling in the global response.
> 3. Ideally, Assumption 1 consistently holds as the cosine similarity of a gradient vector with itself is always 1. In our specific context, confirming Assumption 1 requires precise computation of $\nabla L_j^i\left(\Psi^i\left(t_1\right)\right)$, a domain less explored for neural networks, to the best of our knowledge. Nonetheless, even though the exact verification of this assumption is intractable, the empirical results align with widely accepted observations in ML4CO. Thus, we assert that it holds empirically.
> 4. In Fig 3(a), the positive or negative signs merely signify the direction of influence on other tasks and it's reasonable to have negative transfers within the same COP types if their scales are distinct too much.
>
>     As for the positive entries in the off-diagonals, Fig 3(b) provides a more detailed illustration, and we have discuss this comprehensively in Section 4.2: "*Notably, tasks belonging to the same type of COP are highly influential towards each other due to the large variance of their influence values. Conversely, influences between different types of COPs are negligible, evident from the influence values being concentrated around 0.*"
> 5. In this work, we concentrate on the tasks at hand, and no zero-shot transfers are contemplated. The generalization ability of ML4CO methods poses a persistent challenge, often yielding unsatisfactory results when applied to unseen scenarios without additional efforts. The model derived by our method can serve as a pre-trained model for potential integration with fine-tuning techniques. This avenue holds promise for future exploration but falls outside the scope of the current paper.

---

> > ### Comment · Reviewer_ezHK · 2023-11-21
> > **Discussion**
> >
> > Thank you for replying to my questions.
> >
> > - Regarding this claim "Traditional MTL methods, involving simultaneous training of all tasks, prove to be time-consuming and memory-intensive. As evidenced by the comparisons in Table 3, our approach, which selectively trains one task at each time slot, demonstrates efficiency and effectiveness":  Appendix I seems to suggest that the gain in standard MTL scenarios is not as clear as the gain in COP scenarios. What do you think is specific to COP that makes your strategy especially successful here?
> >
> > - Regarding the work by Fifty et al. (2021), I do not agree with your classification as it does not use any extra information to train the MTL system, and thus is comparable to your method. A fair comparison would be to allow your total training epochs to be the same as the sum of their online+offline epochs.

---

> > > ### Author Response · Authors · 2023-11-22
> > >
> > > Thank you for your feedback.
> > > - The claim that "Traditional MTL methods, involving simultaneous training of all tasks, prove to be time-consuming and memory-intensive" highlights a fundamental limitation of traditional MTL approaches when applied to COPs. These methods often encounter memory constraints due to the collection of information from all tasks, such as gradients. Consequently, our proposed approach offers a distinct advantage by adopting a different training paradigm that prioritizes efficiency and effectiveness. This advantage is clearly demonstrated in the comparison with the random policy, as discussed in the global response.
> > >
> > > - As you mentioned, the work by Fifty et al. (2021) involves both an online and offline phase, while our method and the considered baselines focus on end-to-end online training. The disparity in training methodologies and the challenge of balancing resource allocation between offline and online training make a direct comparison difficult. The reason for referencing Fifty et al. (2021) in our work is that we draw inspiration from their concept of leveraging gradients' effects on the loss function, which influenced the loss decomposition approach adopted in our work.

---

### Official Review · Reviewer_48WT · 2023-10-31

**Soundness:** 3 good
**Presentation:** 2 fair
**Contribution:** 2 fair
**Rating:** 3
**Confidence:** 3

**Summary:**

This paper studies how to efficiently train a multi-task neural solver for multiple combinatorial optimization problems (COPs). They use the gradient information to construct a measure of the similarity of tasks, which is in turn used in the construction of rewards for a multi-armed bandit sampler used to balance the training of multiple tasks. Extensive simulation results are presented to validate the performance of the proposed algorithm over single-task learning and multi-task learning baselines.

**Strengths:**

# Origionality
- Applying MTL to solve multiple COPs seems to be novel.
# Quality
- The experiments are extensive and in detail.
# Clarity
- The paper is in general well-written and smooth to follow, with the exception that some notation in Section 3 are a bit messy.
# Siginificance
- This paper might be of interest to researchers in MTL as it successfully solve multiple COPs with different scales.

**Weaknesses:**

My main concerns are about the novelty and significance of this work.
- Although applying MTL to COPs is less studied, the methodologies presented in this paper (e.g. similarity measure based on gradient information, MAB algorithms) have been well developed. This paper likely attempts to combine them within the context of COPs. In fact, extracting similarity measures using gradient information has been considered in the literature (Wang et al., 2020; Yu et al., 2020). Applying bandit algorithms in MTL has also been studied before (Mao et al., 2021).
- This is a pure experimental paper without theoretical guarantees. I would appreciate some performance guarantees based on the well-developed theoretical results in MAB problems.
- In terms of the performance of the proposed method shown in the experiments, I don't think it "achieves much higher overall performance, ..., compared to standard training schedules". First, in the comparison under the same training budget, each COP naively receives an equivalent budget of $B/4$. Yet, $STL_{avg}$ still archives the best gap in many tasks (Table 2). Second, for comparison under the same training epochs, although I agree this is not a fair comparison, the performance is just equivalent to 100-200 epochs of STL, not significantly larger than the naive calculation $1000/12 \approx 83$ epochs. I would appreciate it if the authors could provide more evidence about the superiority of the proposed algorithm.

**Questions:**

- It seems that the correlation of training the same COP with different scales might be negative, e.g. training TSP-20 pm TSP-100. This is very counterintuitive to me. Could the author explain why this happens?
- I'm curious whether the bandit algorithm is really bringing significant improvement in balancing the training. If we just randomly sample the tasks or naively assign them weights according to the scales of the tasks, what would be the performance?

---

> ### Author Response · Authors · 2023-11-18
>
> Thank you for the efforts on proposing these comments and questions.
> # Response to the weaknesses
> - This work represents a pioneering effort in addressing multiple COPs simultaneously using a unified model. It goes beyond mere amalgamation of existing methodologies within the COP context, and our distinct contributions can be elucidated as follows:
>     1. About the similarity measure: **a)** While (Wang et al. (2020) and Yu et al.) heuristically propose a gradient-based similarity measure and discuss its rationale, our approach starts from a foundational perspective through loss decomposition (Proposition 1). This establishes a theoretical foundation for the heuristics employed by (Wang et al. and Yu et al.). **b)** The similarity measure serves as a pivotal assumption (Assumption 1) that facilitates the design of our proposed algorithm, circumventing the need for gradient estimation in the mean value theorem (Equation (8) in the revised paper): $\nabla^TL^i_j({\Psi}^i(t_1)) = \sum_{t=t_1}^{t_2}\eta_t\mathbb{1}(a_t=T^i_j) \nabla L^i_j(\Theta^i(t))$
>     2. Concerning the application of bandit algorithms in MTL: Both theoretically and practically, our methodology and training paradigm differ from Bandit-MTL (Mao et al., 2021), which necessitates collecting loss information from all tasks and training them collectively at each time slot. In contrast, our approach involves selecting and training a single task using the bandit algorithm, resulting in superior performance (refer to Table 2) and enhanced efficiency (refer to Table 3).
> - It is essential to acknowledge that bandit algorithms thrive under specific assumptions about rewards, environmental properties, and more. The theoretical guarantees sought by the reviewer can be extrapolated if our proposed gradient-based reward and the environment, as simulated by the training process of neural networks, adhere to the corresponding assumptions. Although investigating the correlation between parameter variations induced by training neural networks and the bandit setting's environmental aspects is an intriguing avenue, we believe it surpasses the scope of this work.
> - For the experiments,
>     1. When considering the same training budget, the primary focus should be on the overall performance across all tasks, indicated by the Avg. Gap in Table 2, where our method excels. Notably, the superior performance achieved by $\text{STL}{\text{avg.}}$ is confined to relatively straightforward tasks, such as COPs with smaller problem scales. However, this approach tends to disproportionately allocate resources to these easier tasks, resulting in inadequate training for more challenging ones. A comparison between $\text{STL}{\text{avg.}}$ and $\text{STL}_{\text{bal.}}$ reveals that the latter achieves better overall performance (Avg. Gap) through more judicious resource allocation.
>     2. In the context of the same training epochs, it is crucial to emphasize that the comparison involves a single model generated by our method after 1000 epochs versus 12 models produced by the corresponding STL training scheme, each trained for 1000 epochs. Our objective is to underscore the remarkable generalization ability and efficiency of the neural solver obtained through our method, manifested in the following aspects: **a)** Our method yields a model capable of solving diverse COPs, a feat challenging for models trained on specific COPs due to disparate network structures and distinct state transition logic. **b)** Considering performance on the same kind of COPs, our method outperforms in CVRP, OP, and KP, as evidenced by the "Mean Results" in comparison with COP-small, COP-median, and COP-large. This advantage is attributed to the fact that models trained on one problem scale may perform badly on different scales. Even for TSP, considered the simplest among the four problems, our method maintains competitive performance. **c)** The training time for our method, averaging 0.56 minutes per epoch (Table 3), is significantly shorter than the most resource-intensive alternative (see Table 1), yet it effectively solves diverse COPs.
> # Response to the questions
> - This observed phenomenon aligns with the considerations surrounding the generalization ability of neural solvers for COPs and has been documented in prior studies, as illustrated in Fig. 5 in (Kool et al., 2019)  and Fig. 3 in [1]. Our empirical validation in Fig. 3 within this paper further substantiates this observation. Within the ML4CO community, the challenge of effectively generalizing a model trained on one problem scale to another remains an open question. This knowledge gap motivated our exploration and proposal of MTL methods, specifically designed to train neural solvers capable of handling COPs across various problem scales.
> - Thanks for the advice on the ablation study, we show the results of random policy in the global response.
>
> [1] Learning the travelling salesperson problem requires rethinking generalization

---

> > ### Comment · Reviewer_48WT · 2023-11-22
> >
> > I appreciate the authors' detailed responses and additional experiments on the random policy. Still, I don't think the use of bandit algorithms is well-justified. From a theoretical side, it's hard to determine whether the proposed gradient-based reward and the environment adhere to the assumptions of common bandit algorithms. From a practical side, the improvement from baselines to the random policy is more significant than the improvement from the random policy to the bandit algorithm. It's unclear whether the improvements are due to the encoder structure, the increased network size, or any other factors.
> >
> > Moreover, the authors didn't directly respond to my concern "The performance is just equivalent to 100-200 epochs of STL, not significantly larger than the naive calculation $1000/12\approx 83$ epochs". Here, the total number of train epochs for both approaches is 1000, i.e. 1000 epochs for the proposed bandit method and 1000/12*12 epochs for training on each task separately.
> >
> > Also, I read other reviews and their responses. Reviewer ezHK has the same question (Question 4 there) as mine Question 1, but I don't see the logic in the response "It's reasonable to have negative transfers within the same COP types if their scales are distinct too much". I also agree with other reviewers that Assumption 1 needs to be further justified and discussed.
> >
> > Therefore, I tend to keep my rating at this stage.

---

> > > ### Author Response · Authors · 2023-11-22
> > >
> > > ## About the comments "I don't think the use of bandit algorithms is well-justified"
> > > - In addressing the concern regarding the theoretical justification for employing bandit algorithms, we assert that the bandit algorithms utilized in our study rely on general and reasonably weak assumptions. Specifically, they accommodate bounded stochastic rewards, as exemplified by Exp3, enabling them to effectively handle unknown reward distributions, noisy feedback, and dynamic environments. These characteristics align with the nature of the reward structure proposed in our work and the environment governed by neural network training. To facilitate a more comprehensive understanding, we welcome an elaboration on the specific aspects of theoretical misalignment that the reviewer finds concerning.
> > > - Regarding the practical perspective, while it is acknowledged that the improvements achieved by our method over the random policy may not be as pronounced as those observed over MTL methods, it is imperative to highlight that these advancements are **statistically significant**. The reported results are derived from an extensive average over 10,000 instances, reinforcing the reliability and significance of the observed improvements. Furthermore, we emphasize the novelty of our training paradigm tailored for COPs, constituting a distinctive contribution to the field. It is crucial to note that all compared methods, including MTL baselines, the random policy, and our proposed approach, share identical network structures, model sizes, hyperparameters, and training devices. This meticulous alignment ensures a fair comparison, thereby attributing the observed improvements conclusively to the altered training paradigm and the application of bandit algorithms.
> > >
> > > ## About the comments "The performance is just equivalent to 100-200 epochs of STL, not significantly larger than the naive calculation 83 epochs"
> > > - We reevaluate the results of our method (trained 1000 epochs) and STL(each trained 100 epochs) and show the metrics w.r.t. the optimality gap as follows:
> > >
> > >     |          | TSP20  | TSP50  | TSP100 | CVRP20 | CVRP50 | CVRP100 | OP20   | OP50   | OP100  | KP50   | KP100  | KP200  |  Avg |
> > >     |-------|--------|--------|--------|--------|--------|---------|--------|--------|--------|--------|--------|--------|--------|
> > >     |  STL  | **0.011**  | 0.244  | 1.578  | 0.465  | 1.706  | 3.194   | **-1.133** | 0.781  | 2.898  | **0.026**  | 0.013  | 0.01237  | 0.816|
> > >     |   Ours  | 0.019  | **0.202**  | **1.086**  | **0.348**  | **1.284**  | **2.362**   | -1.114 | **0.224**  | **1.277**  | 0.030  | **0.012**  | **0.01236**  | **0.478**|
> > >
> > >     Upon review, our method  (trained 1000 epochs)  consistently outperforms STL (trained 100x12=1200 epochs) across most tasks, with exceptions noted in TSP20, OP20, and KP50. To enhance clarity, we commit to revisiting Figure 2, refining the presentation to provide a more explicit and informative comparison in the upcoming version. We appreciate the diligence of the reviewer in identifying this point, and we believe this additional analysis strengthens the justification for the effectiveness of our proposed bandit method.
> > >
> > > ## About the claim "It's reasonable to have negative transfers within the same COP types if their scales are distinct too much".
> > > - In addressing the shared concern raised by both you and Reviewer ezHK regarding the explanation provided for Question 1 (Question 4 for ezHK), we have quoted the conclusion derived from extensive experiments (Fig. 5 in [1] and Fig. 3 in [2]), which further motivates our work in some sense. We believe our analysis, as well as these impactful works, are sufficient to demonstrate the common points of negative transfer in the ML4CO community. By now, we can only obtain evidence supporting our claim, either empirically, or from previous arts. We wonder if the reviewer can provide any evidence that such a claim is inappropriate.
> > >
> > >     [1]. Kool W, Van Hoof H, Welling M. Attention, learn to solve routing problems![J]. arXiv preprint arXiv:1803.08475, 2018.
> > >
> > >     [2]. Joshi C K, Cappart Q, Rousseau L M, et al. Learning the travelling salesperson problem requires rethinking generalization[J]. arXiv preprint arXiv:2006.07054, 2020.
> > >
> > > - About Assumption 1, we would like to draw your attention to the comprehensive explanations provided in response to Reviewer ezHK's Question 3 and Reviewer 7uDV's Weakness 2. These discussions delve into both theoretical and practical perspectives, aiming to elucidate the rationale behind Assumption 1. We remain confident that these detailed explanations sufficiently address the concerns raised and provide a robust foundation for the validity of Assumption 1.

---

### Author Response · Authors · 2023-11-18
**Results of Random Policy**

We show the short version for the comparison with random policy as follows. The full version of the results can be found in Table 2 in the paper.

Small Budget:
| Method | TSP20 | TSP50 | TSP100 | CVRP20 | CVRP50 | CVRP100 | OP20 | OP50 | OP100 | KP50 | KP100 | KP200 | Avg. Gap |
|-|-|-|-|-|-|-|-|-|-|-|-|-|-|
| $\text{STL}_{\text{avg.}}$ | **0.009%** | 0.346% | 3.934% | 0.465% | 2.292% | 5.899% | -1.075% | 1.291% | 5.674% | **0.029%** | 0.015% | 0.017% | 1.575% |
| $\text{STL}_{\text{bal.}}$ | 0.019% | 0.346% | 2.967% | 0.599% | 2.292% | 4.774% | -0.973% | 1.291% | 4.771% | 0.033% | 0.015% | 0.016% | 1.346% |
| UW | 0.036% | 0.394% | 1.905% | 0.451% | 1.667% | 3.291% | -0.562% | 1.776% | 3.989% | 0.039% | 0.016% | 0.022% | 1.085% |
| Random | 0.041% | 0.402% | 1.975% | 0.489% | 1.797% | 3.298% | -0.998% | 0.794% | 2.488% | 0.032% | 0.014% | 0.015% | 0.862% |
| Ours | 0.030% | **0.297%** | **1.687%** | **0.422%** | **1.554%** | **2.861%** | **-1.081%** | **0.533%** | **2.153%** | 0.031% | **0.014%** | **0.014%** | **0.710%** |

Median Budget:
| Method | TSP20 | TSP50 | TSP100 | CVRP20 | CVRP50 | CVRP100 | OP20 | OP50 | OP100 | KP50 | KP100 | KP200 | Avg. Gap |
|-|-|-|-|-|-|-|-|-|-|-|-|-|-|
| $\text{STL}_{\text{avg.}}$ | **0.005%** | **0.183%** | 2.237% | 0.379% | 1.667% | 4.137% | **-1.156%** | 0.674% | 3.670% | **0.025%** | 0.013% | 0.015% | 0.988% |
| $\text{STL}_{\text{bal.}}$ | 0.008% | 0.183% | 1.656% | 0.447% | 1.667% | 3.460% | -1.118% | 0.674% | 2.563% | 0.027% | 0.013% | 0.013% | 0.800% |
| UW | 0.036% | 0.231% | 1.426% | 0.373% | 1.540% | 3.032% | -0.855% | 0.802% | 3.108% | 0.033% | 0.013% | 0.022% | 0.813% |
| Random | 0.025% | 0.252% | 1.310% | 0.371% | 1.438% | 2.608% | -1.099% | 0.339% | 1.583% | 0.029% | 0.013% | 0.014% | 0.574% |
| Ours | 0.019% | 0.202% | **1.086%** | **0.348%** | **1.284%** | **2.362%** | -1.114% | **0.224%** | **1.277%** | 0.030% | **0.012%** | **0.012%** | **0.478%** |

Large Budget:
| Method | TSP20 | TSP50 | TSP100 | CVRP20 | CVRP50 | CVRP100 | OP20 | OP50 | OP100 | KP50 | KP100 | KP200 | Avg. Gap |
|-|-|-|-|-|-|-|-|-|-|-|-|-|-|
| $\text{STL}_{\text{avg.}}$ | **0.002%** | **0.114%** | 1.296% | **0.282%** | 1.276% | 3.071% | **-1.223%** | 0.253% | 2.087% | **0.018%** | 0.011% | 0.014% | 0.600% |
| $\text{STL}_{\text{bal.}}$ | 0.005% | 0.114% | 1.024% | 0.367% | 1.276% | 2.601% | -1.175% | 0.253% | 1.609% | 0.024% | 0.011% | 0.012% | 0.510% |
| UW | 0.025% | 0.195% | 1.230% | 0.347% | 1.399% | 2.718% | -0.855% | 0.588% | 2.022% | 0.028% | 0.013% | 0.015% | 0.644% |
| Random | 0.019% | 0.207% | 1.069% | 0.336% | 1.274% | 2.322% | -1.104% | 0.219% | 1.188% | 0.027% | 0.012% | 0.012% | 0.465% |
| Ours | 0.017% | 0.169% | **0.960%** | 0.335% | **1.218%** | **2.236%** | -1.139% | **0.149%** | **1.129%** | 0.026% | **0.011%** | **0.011%** | **0.427%** |

Interestingly, the random policy outperforms STL and the best-performing MTL baselines in our context, underscoring the positive effects of changing the training paradigm. Furthermore, our proposed method surpasses the random policy, providing evidence of the additional improvements achieved through the integration of the bandit algorithm.

---

### Meta-Review · Area_Chair_vZEL · 2023-12-05

**Metareview:**

This paper applies bandit algorithms to select training tasks for a multi-task neural solver for combinatorial optimization. The setting is novel and interesting. The shortcomings of the paper are:

* **Novelty:** This paper builds heavily on known methods and results. Many empirical results are not surprising. In my opinion, this is not a major issue because the combination of the methods is interesting.

* **Motivation for bandit algorithms:** We are not convinced that bandits are the right approach to solving this problem. This was actually showed in the rebuttal. The fact that a random policy is often better than the original baselines, and close to the bandit algorithm, indicates that something else is going on.

* **Analysis:** None. Therefore, it is not clear why the proposed method performs better. I am surprised that nothing can be done for Exp3, since this algorithm has worst-case guarantees that can be combined with almost anything.

No reviewer supported acceptance of this paper and therefore it is rejected.

**Justification For Why Not Higher Score:**

We are not convinced that bandits are the right approach to solving this problem. The authors provided evidence for this during the rebuttal.

**Justification For Why Not Lower Score:**

N/A

---

### Decision · Program_Chairs · 2024-01-16

Reject